# Spatial self-organization resolves conflicts between individuality and collective migration

X. Fu [1,6], S. Kato[1,7], J. Long[1,2], H.H. Mattingly[1], C. He[3], D.C. Vural [1,8], S.W. Zucker[4,5] & T. Emonet [1,2]

Collective behavior can spontaneously emerge when individuals follow common rules of interaction. However, the behavior of each individual differs due to existing genetic and non-genetic variation within the population. It remains unclear how this individuality is managed to achieve collective behavior. We quantify individuality in bands of clonal *Escherichia coli* cells that migrate collectively along a channel by following a self-generated gradient of attractant. We discover that despite substantial differences in individual chemotactic abilities, the cells are able to migrate as a coherent group by spontaneously sorting themselves within the moving band. This sorting mechanism ensures that differences between individual chemotactic abilities are compensated by differences in the local steepness of the traveling gradient each individual must navigate, and determines the minimum performance required to travel with the band. By resolving conflicts between individuality and collective migration, this mechanism enables populations to maintain advantageous diversity while on the move.

[1] Department of Molecular, Cellular, and Developmental Biology, Yale University, New Haven, CT 06520, USA. [2] Department of Physics, Yale University, New Haven, CT 06520, USA. [3] Institute of Synthetic Biology, Shenzhen Institutes of Advanced Technology, Chinese Academy of Sciences, Shenzhen 518055, China. [4] Department of Computer Science, Yale University, New Haven, CT 06520, USA. [5] Department of Biomedical Engineering, Yale University, New Haven, CT 06520, USA. [6]Present address: Institute of Synthetic Biology, Shenzhen Institutes of Advanced Technology, Chinese Academy of Sciences, Shenzhen 518055, China. [7]Present address: Department of Molecular Biotechnology, Graduate School of Advanced Sciences of Matter, Hiroshima University, Higashi-Hiroshima, Hiroshima 739-8530, Japan. [8]Present address: Department of Physics, University of Notre Dame, Notre Dame, IN 46556, USA. These authors contributed equally: X. Fu and S. Kato.  Correspondence and requests for materials should be addressed to T.E. (email: thierry.emonet@yale.edu)

Cells and larger organisms exhibit collective behaviors that are often advantageous to the participating individuals[1,2]. Many such collective behaviors dynamically emerge when a large number of individuals follow the same rules to interact with each other and the environment[3,4]. Prominent examples are bird flocks[4–6] and the collective migration of bacteria along channels[7–11] and on agar plates[8,9,12]. At the same time, phenotypic differences among even genetically identical individuals are a ubiquitous feature of biology[13]. Phenotypic diversity can lead to useful leader–follower structures within a traveling group. For example, in migrating neural crest cells and in fish shoals, many organisms may follow a few more informed individuals[14]. In microbial communities, maintaining diversity in the population can enable bet-hedging strategies to survive uncertain environments and resolve trade-offs[13,15–18]. However, heterogeneity can also be disruptive, as is the case in simulated swarms where non-aligners tend to be purged from the swarm[19]. This raises a dilemma: although phenotypic diversity provides advantages, it also tends to reduce coordination.

One of the simplest cases of collective behavior is exhibited by bacteria: clonal populations of motile *Escherichia coli* cells collectively migrate when placed at high density at the bottom of a tube filled with nutrients[8–11]. This collective behavior is mediated by the well-characterized chemotaxis system[20], which enables the bacteria to follow chemical gradients, in this case generated by their consumption of attractant present in the medium (Fig. 1a). However, populations of *E. coli* exhibit substantial cell-to-cell variability in their swimming phenotypes[13,21] and hence chemotactic abilities, even when all cells are genetically identical[22]. How bacterial populations manage phenotypic heterogeneity to still allow coordinated collective migration remains largely unknown, mainly because of the difficulties in measuring cellular behavior at both the collective and the individual levels in the same experiment[21].

Although the migration of traveling waves or "bands" of bacteria has also served as a classic model for the theoretical study of emergent phenomena and pattern formation in biology[10,23], the effect of non-genetic diversity on this process has scarcely been examined. Previous studies examined how two populations may travel together[24,25]; however, it was assumed that within each population all of the individuals were identical. The mechanisms by which a continuum of phenotypes can achieve coherent migration have not been investigated.

Here we used a microfluidic system that enables precise quantitative measurements at the individual and collective scales to study the interplay of diversity and collective bacterial migration. Our central finding is that within the traveling band, cells spontaneously sort themselves such that their chemotactic abilities are matched to the local gradient steepness, enabling diverse cells to travel together with the same drift speed. Extending the classic Keller–Segel model of traveling bands to account for diversity predicts this spatial sorting and qualitatively recapitulates the experimental results. Our second finding is a novel mechanism that reduces the rate at which cells fall off the back of the band: when attractant consumption depends on local oxygen, oxygen limitation in the center of the band increases the gradient of attractant at the back, helping cells there keep up. Together, these two mechanisms enable populations of bacteria to maintain diversity while migrating as a group.

## Results

### Cells of diverse chemotactic abilities migrate as a group.
To quantify collective behavior and diversity in the same experiment, we designed a microfluidic device consisting of a long channel to observe the traveling band[11], followed by a large chamber to quantify the distribution of phenotypes in the band (Fig. 1b and Supplementary Fig. 1). Approximately $2 \times 10^5$ clonal *E. coli* cells grown in M9 glycerol medium (M9 salts, glycerol, and casamino acids; Methods) were introduced with fresh medium into the device and concentrated at the end of the channel by centrifugation (Methods). Following centrifugation, sequential bands of cells collectively migrated along the channel at different but nearly constant speeds (Fig. 1c), presumably consuming different compounds within the undefined media, as demonstrated in early studies[8].

*E. coli* cells navigate by alternating straight "runs" with "tumbles" that randomly reorient their swimming direction (Fig. 1d). By transiently suppressing tumbles whenever attractant signal increases, they perform a biased random walk that allows them to move toward higher concentrations of attractant[20]. In the absence of a gradient, the fraction of time a cell spends tumbling —its tumble bias (TB)—remains approximately constant and therefore can be used as a quantitative measure of the phenotype of the cell. Importantly, using the same strain and microfluidic channel depth, we previously demonstrated that the TB is a strong determinant of chemotactic performance in liquid: lower TB cells drift significantly faster up a static gradient than higher TB cells[22]. To quantify the distribution of phenotypes in the isogenic population that was introduced in the device, a low density of cells was loaded into the microfluidic device without centrifugation and individual cells were tracked to determine their TB, as previously described[21,22]. TB was broadly distributed in the population with some cells tumbling < 10% of the time (i.e., TB < 0.1) and others > 50% of the time (Fig. 1e black), consistent with previous studies[13,21,22]. Given the functional consequences of this non-genetic diversity, how can the same population of cells migrate together as a coordinated group, as shown in Fig. 1c?

To answer this question, we first considered whether all phenotypes or only a subset of them traveled in each band. We used pressure valves to capture one band of cells at a time in the wide chamber of our device (Fig. 1b and Supplementary Fig. 1b). After trapping cells in the wide chamber, it was perfused with fresh media to homogenize the environment and dilute the cell density. We verified that perfusion of the wide chamber did not affect the distribution of TBs (Supplementary Fig. 3). Dilution enabled us to track individual cells. Homogenization ensured that cells had adapted back to a uniform environment and were not responding to an attractant gradient when we measured their TBs. The distribution of TB was shifted toward lower TB in both traveling bands compared with the original distribution (Fig. 1e), suggesting that it was more difficult for high TB cells to participate in collective migration. Selection against high TB cells was stronger in the faster band (Fig. 1e, red) than in the slower one (Fig. 1e, yellow). Cell density and number also varied between the two bands (Fig. 1c), suggesting that there were interdependencies between the speed of the group, its size, and the diversity of the individuals able to migrate with the group. We periodically tracked cells after they were trapped and diluted in the wide chamber and found that the original TB distribution was recovered after growth (Fig. 1f). Thus, selection of low TB cells by the collective migration was not due to genetic heterogeneity. In addition, it is unlikely that cell growth affected the TB selection while cells were traveling in a band, because the duration of the experiment (30 min) was shorter than the cell doubling time (~ 55 min, Supplementary Fig. 4).

### Collective migration selects against high TB cells.
To determine the relationship between the number of cells in the band, the band speed, and diversity, we switched from casamino acids to a defined M9 glycerol buffer containing aspartate (Asp) as the only limited chemoattractant (Methods). In this condition, a

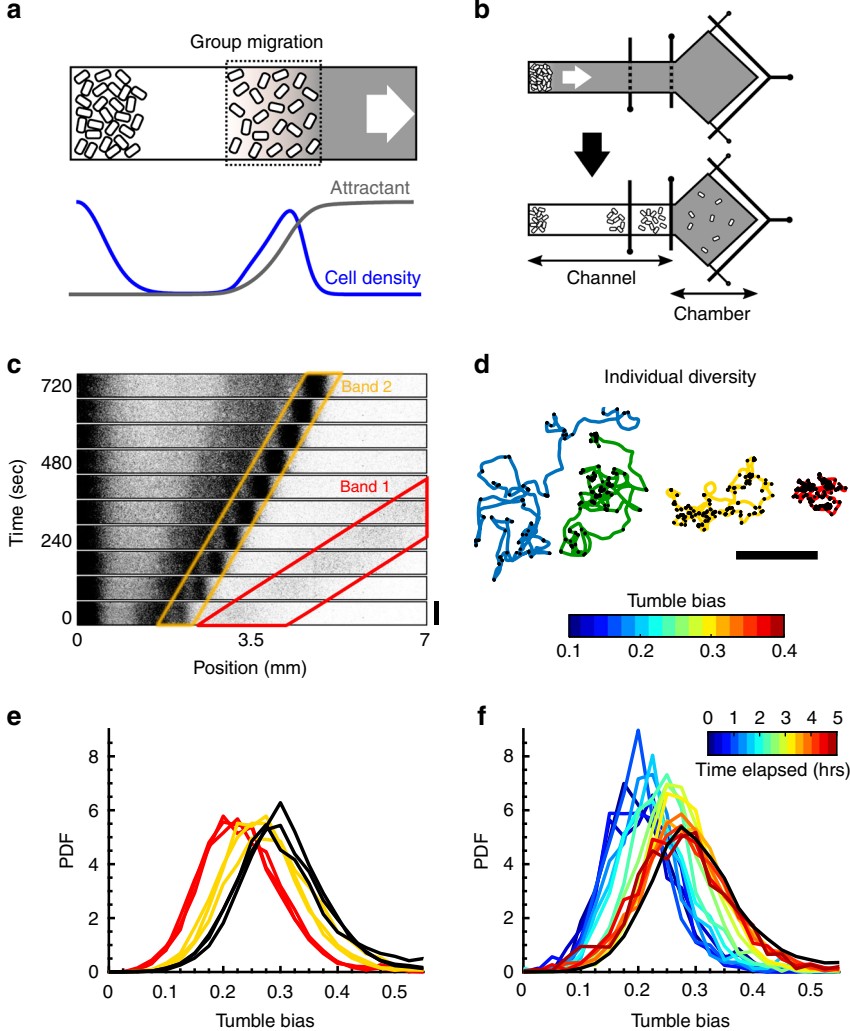

**Fig. 1** Collective migration of a phenotypically-diverse clonal population. **a** When concentrated at the bottom of a nutrient channel, motile *E. coli* cells emerge from the high cell density region and travel in bands along the channel by following gradients of attractant produced by their consumption. **b** Microfluidic device used to quantify the band migration and the phenotypic diversity within the band. Control gates along the channel (black vertical lines) are initially open (top) and later closed to capture different bands of cells in the observation chamber (bottom), where single cells are tracked to quantify the distribution of phenotypes within the band (Supplementary Fig. 1). **c** Time-lapse imaging of *E. coli* cells expressing the fluorescent protein mRFP1 showing the collective migration of bands in M9 glycerol medium (M9 salts, glycerol, and casamino acids; Methods). In this undefined medium, several bands emerge that travel at different speeds (red: 0.68 mm min$^{-1}$, yellow: 0.23 mm min$^{-1}$)[8]. We verified that labeling cells did not affect band speeds nor tumble bias distributions (Supplementary Fig. 2). Scale bar, 0.6 mm. **d** The tumble bias (color)—average probability to tumble—of individual cells was quantified by tracking a cell for 2 min in a uniform environment (no gradient) and detecting tumbles (black dots) as previously described[21, 22]. Scale bar, 200 μm. **e** Collective migration selects against high TB cells. Tumble bias distributions from the first (red) and second (yellow) bands (*n* = 3), and from the population that was introduced in the device (black; *n* = 3). **f** The tumble bias distribution of the cells in the first wave (red in **e**) gradually shifted back toward the original distribution (black) during growth in the perfused chamber. Fresh M9 glycerol medium was supplied every 30 min. The TB distribution was measured every 20 min

single band formed (Fig. 2a) and its speed could be tuned by changing the concentration of supplemented Asp[12] (Fig. 2b). To measure band speed and density, cells expressing mRFP1 were mixed with unlabeled cells at ratios of 1:20, 1:50, or 1:100 (for 50, 100, and 200 μM Asp, respectively), and their positions were detected at various time points (Fig. 2a and Methods[11,26]). Band speed, number of cells in the band, and density profiles were stable over time with a slow decay due to cells falling off at the back of the band (Fig. 2b, c, d and Supplementary Fig. 5). Although there were variations across experimental replicates due to variations in the number of cells introduced in the device, the relationship between Asp concentration and band parameters was consistent. Specifically, as the concentration of Asp increased, the speed of the band decreased (Fig. 2b), the number and peak

density of cells in the band increased (Fig. 2c, d), and the distribution of TB within the band shifted toward higher TB (Fig. 2e). In general, collective migration selected against high TB cells, with selection being stronger in faster bands (Fig. 2e, f). It is noteworthy, however, that diversity was not eliminated—all bands still exhibited a range of TBs. Thus, although collective migration selected against high TB cells, it was still possible for a diverse group to travel together.

**Extending the Keller–Segel model to account for diversity**. To better understand how collective migration selects TB, we extended the classic Keller–Segel mathematical model describing traveling bands of bacteria[10] to include phenotypic diversity (Methods Eqs. 1–3). In this model, cells consume the diffusible

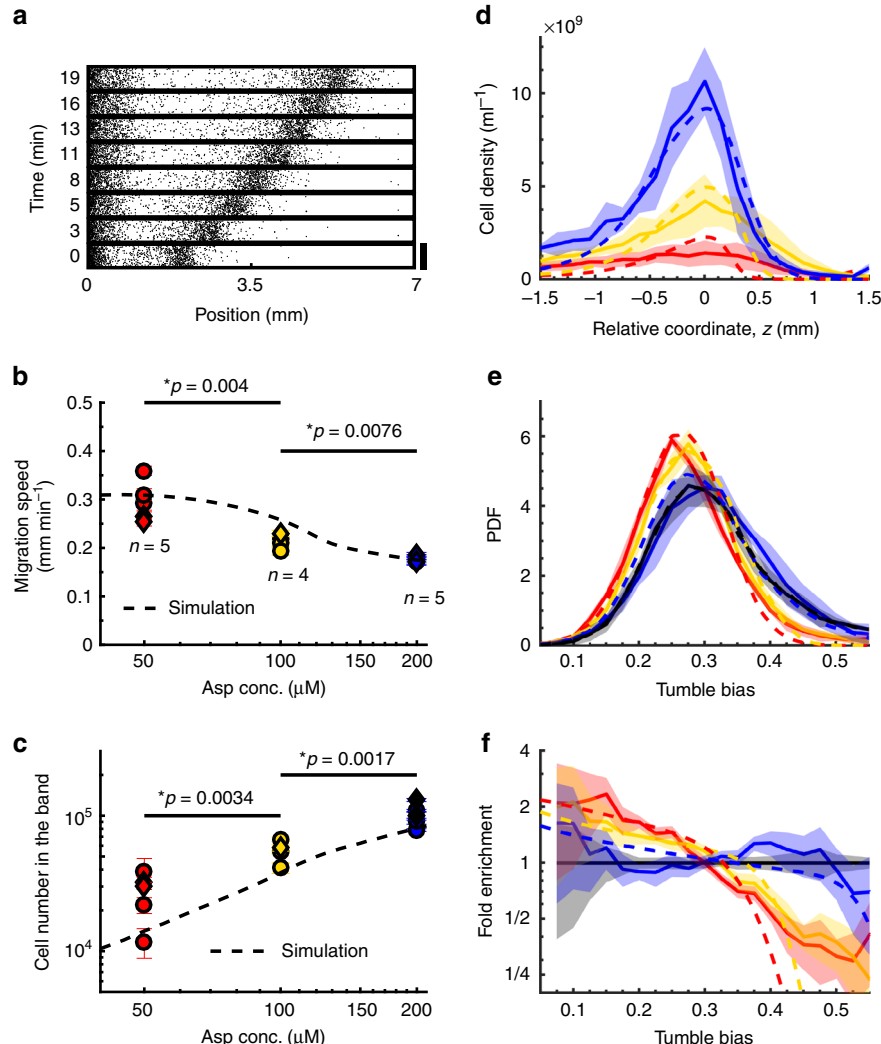

**Fig. 2** Relationship between band speed, density, and phenotypic diversity. **a** Time-lapse coordinates of cells (black dots) traveling in M9 glycerol buffer (Methods) with 200 μM aspartate. Only one band forms in each experiment. Cells (1:100) were labeled with mRFP1 and their coordinates detected (Methods). Scale bar, 0.6 mm. In remaining panels, colors are aspartate concentration in the buffer: 200 μM (blue; 1:100 cells labeled), 100 μM (yellow; 1:50 cells labeled), 50 μM (red; 1:20 cells labeled) (Methods). **b** Band speed decreased with aspartate concentration in the buffer. Circles: experiments in which tumble bias was measured, used in **e**, **f**. Diamonds: tumble bias was not measured. Dashed: simulations (Methods and Supplementary Fig. 5). $p$-value: one-sided, two-sample $t$-test assuming unequal variances, $t$-value = 4.1, df = 5.4 between 50 μM and 100 μM, $t$-value = 4.2, df = 3.8 between 100 μM and 200 μM. **c** Number of cells traveling in the band increased with aspartate concentration. Circles, diamonds, and dashed: same as in **b**. $p$-value: one-sided, two-sample $t$-test assuming unequal variances, $t$-value = −3.9, df = 6.5 between 50 μM and 100 μM, $t$-value = −4.6, df = 6.4 between 100 μM and 200 μM. **d** Cell density profiles averaged over time and experiments (Supplementary Fig. 5). Line: average over $n = 5$ (red), $n = 4$ (yellow), $n = 5$ (blue) replicate experiments. Shade: SD. For each experiment, nine profiles were measured at 1.3 min intervals. Dashed: simulations. **e** Tumble bias distribution of the cells that traveled with the band. Lines: average over experiments; shading: SD; dashed lines: simulations. **f** Ratio between the distribution of tumble bias in the traveling band (Fig. 2e, colored lines) and that of the original population (Fig. 2e, black) quantifies the enrichment of tumble bias. Lower aspartate concentrations enrich more for lower tumble bias. Solid lines: mean of the measurements from experiments; shading: standard error of the mean; dashed: simulations

attractant Asp, which generates a traveling gradient that the cells follow by biasing their random walk (Fig. 3a). Faster consumption, more cells, or less Asp all lead to a faster traveling gradient[10]. The motion of a phenotype $i$ depends on two parameters: its effective diffusion coefficient, $\mu_i = \mu(TB_i)$, which results from the cells' random walk, and its chemotactic coefficient, $\chi_i = \chi(TB_i)$, which quantifies how effective that phenotype is at biasing its motion[27–29] to follow the perceived amount of Asp, $f$. Theory and tracking of individual *E. coli* cells swimming in a static gradient of α-methylaspartate (non-metabolizable analog of Asp) have shown that $\mu$ and $\chi$ are decreasing functions of the TB (note that for very low values of TB < ~0.05, $\chi$ increases with TB;

indeed, for TB = 0 the cell is just diffusing and $\chi = 0$; Methods)[21,22]. Thus, in a gradient of attractant, cells with higher TB do not diffuse as much and climb slower than cells with lower TB. The dependency of $f$ on Asp concentration has been characterized as well[30]. Moreover, we conducted high-performance liquid chromtography (HPLC) experiments to verify that the chemotaxis response to Asp dominated that to amino acids secreted as byproducts of Asp metabolism. Experiments conducted with mutants lacking the oxygen receptor *aer* or both *aer* and *tsr* indicated that aerotaxis was not essential and Tar response to Asp was sufficient for band migration[31] (Methods and Supplementary Fig. 6).

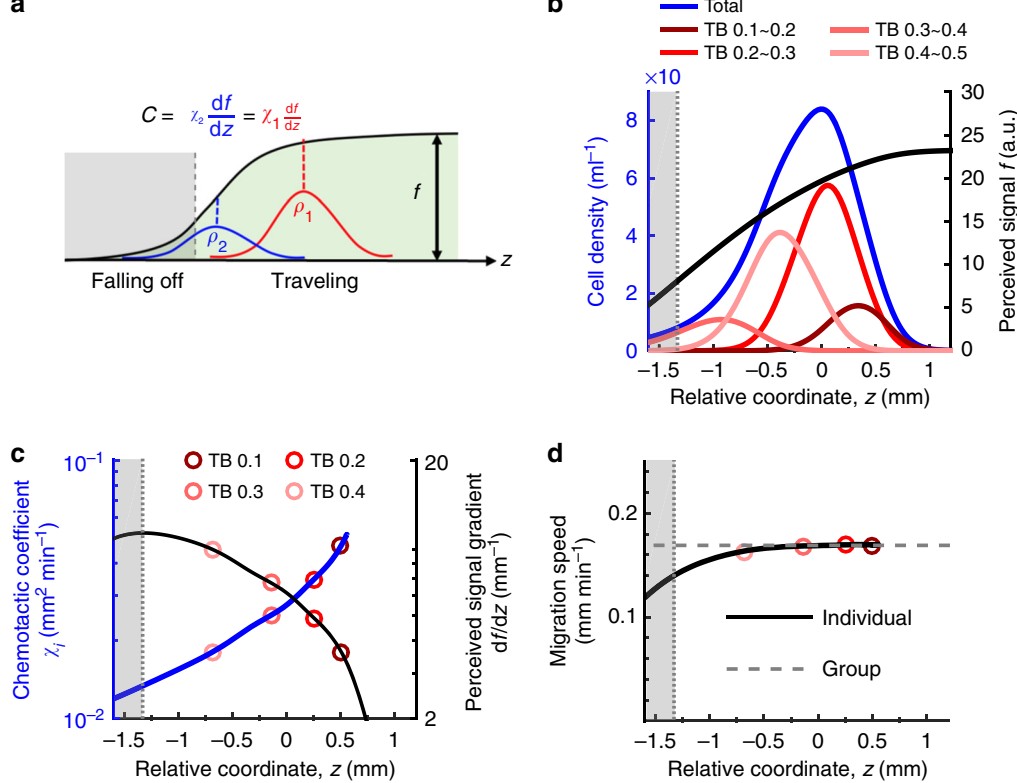

**Fig. 3** Mathematical modeling predicts a mechanism for consistent collective migration of diverse phenotypes. **a** Collective migration of diverse phenotypes at the same speed is made possible by the spontaneous spatial ordering of individual phenotypes within the band such that each individual's chemotactic ability is matched to the local gradient steepness $\frac{df}{dz}$. The proportion of better performers (larger $\chi$, lower TB; red) should be enriched where the gradient is shallower (front), whereas the proportion of weaker performers (smaller $\chi$, higher TB; blue) should be enriched where the gradient signal is steeper (back). The position where the perceived gradient steepness is maximum (dashed border of the gray regions) determines the highest tumble bias able to travel with the band. Cells in the gray region slowly fall out of the band. **b** Simulated density profiles of cells migrating in 200 μM aspartate (the same simulation is shown in Fig. 2def, blue dashed) show sorting based on tumble bias. **c** Chemotactic coefficient $\chi(z)$ (blue) defined by the phenotype whose density profile peaks at position $z$ and perceived gradient steepness $\frac{df}{dz}$ (black). Red symbols correspond to the location of the peak cell density for individual phenotypes. **d** Spatial sorting enables consistent migration velocity for traveling phenotypes. The migration velocity, $\chi(z)\frac{df}{dz}$, of the phenotype whose density profile peaks at position $z$ gradually decreases toward the back of the band until the gray region is reached. In the gray region the migration velocity falls off more rapidly, preventing the high TB phenotypes located there from staying in the band

To metabolize Asp, *E. coli* consumes oxygen[32]. Introducing a fluorescent oxygen sensor[33] in the M9 glycerol buffer revealed that oxygen availability is reduced in the center of the traveling band where cell density is high (Supplementary Fig. 7a-f). This results in a dependency of the average consumption rate of Asp on cell density (Supplementary Fig. 7h). We modeled this effect such that the Asp consumption rate depended linearly on oxygen concentration and constrained the related parameters by measuring oxygen and Asp consumption rates in batch cultures (Supplementary Fig. 7g). For simplicity, we ignored possible phenotypic diversity in the Asp consumption rate, as well as the possible dependence of the diffusion coefficient on cell density[34], which was found previously to be negligible in similar experiments[11]. We also omitted possible contributions of hydrodynamics[35] and physical interactions between cells, which can become important when bacteria swarm over surfaces[36]. The resulting model (Methods and Table 1) qualitatively reproduced the main features of our experiments, including the dependency on Asp concentration of the band speed (Fig. 2b), cell number and density (Fig. 2c, d), TB distribution (Fig. 2e), phenotypic selection (Fig. 2f), and average Asp consumption rate per cell as a function of cell number in the band (Supplementary Fig. 7h).

**Spatial sorting as a mechanism for inclusive migration**. An important feature of the experiments reproduced by the simulations is the increasing selection against high TB cells as the amount of Asp is reduced (Fig. 2f). What mechanism enables cells with diverse chemotactic abilities to collectively migrate together as one band, and what controls the upper bound on the TB among those able to migrate together? For every phenotype that travels with the band at constant speed $c$, the flux of cells must be approximately invariant in time and equal to the chemotactic flux minus that due to diffusion. Focusing on the partial differential equation for the cell density of phenotype $i$ and switching to the moving reference frame $z = x - ct$ (here $x$ and $t$ are absolute position and time), two predictions emerge (Methods Eqs. 7–8). First, the traveling phenotypes must distribute themselves spatially within the band so that differences in local signal strengths, the slope $\frac{df}{dz}$ of the black line in Fig. 3a, compensate for differences in the chemotactic abilities of each phenotype, $\chi_i$, such that $c = \chi_i \frac{df}{dz}(z_i)$, where $z_i$ is the position of peak density of phenotype $i$ (Fig. 3a). Therefore, this spatial sorting places the better performers (higher $\chi_i$, lower $TB_i$) in the front of the band, where the gradient is shallower and more difficult to follow (Fig. 3a, red), and the weaker performers (lower $\chi_i$, higher $TB_i$) at the back,

where the gradient is steeper (Fig. 3a, blue). Furthermore, a second prediction is that the gradient will reach a maximal steepness (Fig. 3a dashed border of gray zone), determining the weakest phenotype that can travel (lowest $\chi_i$, upper bound on $TB_i$). Thus, we see the interplay between individuals $i$ and the effect of the community on the available resource $f$.

Analysis of simulations confirmed these analytical predictions (Fig. 3b, c, d). The steepness of the perceived signal $\frac{df}{dz}$, which emerges dynamically from the cells' consumption, peaked at the back of the traveling band (low $z$) and decayed toward the front (high $z$) (Fig. 3c black). In contrast, the position $z_i$ of the peak density of phenotype $i$ increased with its chemotaxis coefficient $\chi_i$ (Fig. 3c blue), revealing an ordering of the phenotypes within the migratory band (Fig. 3b, c). Multiplying the two together gave a nearly constant velocity throughout the band, thus providing an explanation of how the various phenotypes might travel together (Fig. 3d). The rightmost points in Fig. 3c (blue line) and Fig. 3d (black line) correspond to the phenotype with the maximum chemotactic coefficient in the band. Ahead of that location, there are no more peaks in the cell density of any phenotype; however, there are cells due to diffusion. At the back of the band, cells with TB higher than the predicted upper bound rapidly fell off of the band (Fig. 3a, d gray zones). Importantly, these predictions emerged from just the dynamics of cell density (Methods Eq. 1) and therefore hold true irrespective of whether oxygen is included in the model.

Comparing simulations with and without the oxygen-dependent consumption rate revealed that the oxygen dependency reduces leakage of the highest-TB cells located at the back of the band (Supplementary Fig. 8b). The higher concentration of oxygen at the back relative to the center of the band locally increases the rate of consumption, and hence the slope of the traveling gradient of Asp, helping the cells there stay longer with the band (Methods Eq. 9). Thus, oxygen dependency has a similar cohesive effect as the secretion of a self-attractant, which also helps reduce the leakage of cells[11,25,37].

**Cells in the band are spatially sorted by TB**. To experimentally test the prediction of spatial sorting in the band by phenotype, we measured the relative position of two populations of cells with different mean TB within the traveling band. The distribution of TB in the population was controlled by manipulating the level of expression of the phosphatase CheZ, which deactivates the chemotaxis response regulator CheY (Supplementary Fig. 9a). We generated multiple populations with different TB distributions of varying mean TB (⟨TB⟩) (Fig. 4a). In each population, we labeled 1 in 50 cells of the same genetic background and induction level by ectopically expressing either mRFP1 or YFP. The inducer was washed away when cells were resuspended in buffer before starting the migration experiment. TB distributions are stable for more than an hour in buffer[21]. When cells from the lowest (red) and highest (cyan) TB distributions were mixed in equal parts and introduced in the device, spontaneous spatial order emerged, with cells from the low (high) TB distribution located at the front (back) of the traveling band (Fig. 4b, c and Supplementary Fig. 9b for replicates). The distance between the peaks of the density profiles of the two populations remained nearly constant over the duration of the experiment, indicating that the two populations traveled together at the same speed (Fig. 4e). We verified that the distance between the peak densities of the two populations was stable in longer experiments (Supplementary Fig. 9d). Mixing populations with closer TB distributions caused the peaks of the two traveling populations to be closer (magenta and green in Fig. 4a, d and Supplementary Fig. 9c for replicates), suggesting that distance between peaks increases with difference in ⟨TB⟩ as a result of spatial sorting. Due to experimental limitations, we could not measure the TB distributions of all four strains in each of the experiments reported in Fig. 4a-e. To better quantify the relationship between peak separation and difference in ⟨TB⟩ value, we instead mixed pairs of populations using only fluorescent strains induced with different levels of anhydrotetracycline (aTc). For each pair, we measured the TB distributions right before loading cells in the device, and then measured the distance between the fluorescence peaks in the resulting traveling band (Fig. 4f). This confirmed that there is a monotonic relationship between peak separation and difference in ⟨TB⟩ values. Therefore, cells of various phenotypes appear to spontaneously sort themselves along the traveling band according to their TB, enabling them to migrate collectively despite phenotypic differences.

**Discussion**

How do organisms maintain collective behavior despite the potential conflicts created by phenotypic diversity among individuals? We studied this question using traveling bands of chemotactic E. coli, which collectively migrate at the same speed despite differences in chemotactic abilities of individuals in the band. Our key result is that spontaneous spatial organization of phenotypes within a traveling band helps resolve the conflicts between phenotypic diversity and collective migration. By matching individual abilities to the local difficulty of the navigation task within the band, this sorting mechanism ensures consistent migration speed across the band. This process also determines the minimum chemotactic performance required to keep up with the band, therefore explaining how diversity can become limited by collective behavior. Thus, the mechanism reported here enables a continuum of phenotypes to migrate coherently.

In the traveling band, there is always a slow leakage of cells off the back of the band because of the finite sensitivity of the cells for the attractant they are chasing[38–41]. High TB cells, in particular, which are localized at the back of the wave, are at risk of falling off. We discovered that this leakage can be reduced (but not eliminated) if the consumption rate of the attractant is lower in the center of the band than at the back, where the consumption rate determines the local gradient steepness and chemotactic drift. In our case, this arises because Asp consumption depends on oxygen, which becomes limited in the center of the band where cell density is high. This mechanism provides an alternative to other mechanisms known to reduce cell leakage, such as the secretion of an attractant by the traveling cells[11,25,37]. Note that the spontaneous sorting mechanism discussed above helps compensate for differences in chemotactic abilities, irrespective of the presence of such auxiliary mechanisms (oxygen or self-attractant).

Traveling bands of bacteria have been studied for decades since Julius Adler's experiments in capillary tubes[8,9]. Adler reported the formation of multiple traveling bands in complex media; we observe the same in casamino acids (Fig. 1) and expect multiple bands to be able to form when multiple consumable attractants are present. Within a migrating band the cells respond to the traveling gradient, some parts of which can be fairly steep. Therefore, we expect the instantaneous TB of an individual cell to be dynamically changing depending on its direction of motion and position within the gradient, as previously reported[11,29]. Here we showed that phenotypic (intrinsic) differences in adapted TB between cells contribute significantly to spatial structure within the traveling band. In future studies, we will separate the contributions of phenotypic and dynamic diversity to group structure. It will also be interesting to examine the contribution of initial conditions, dimensionality, and growth (necessary to

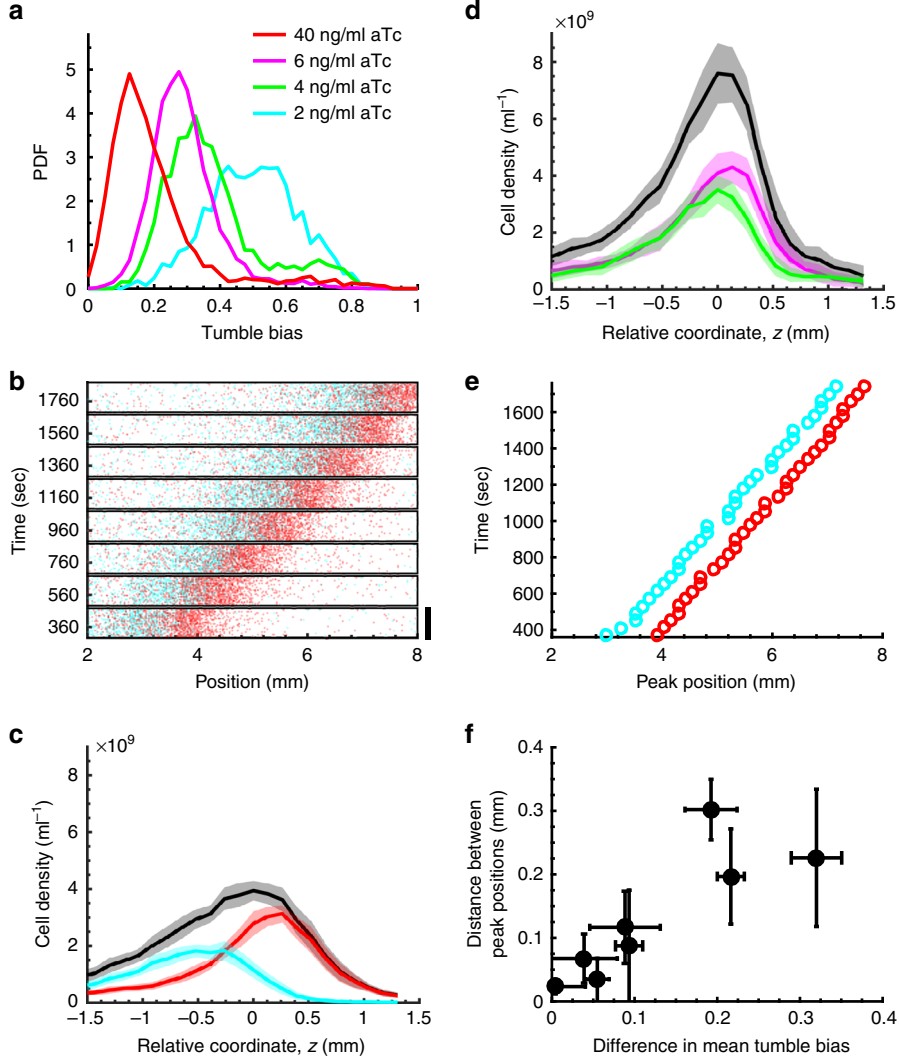

**Fig. 4** Phenotypes spontaneously order themselves along the traveling band according to tumble bias. **a** The distribution of tumble bias in the population can be controlled by manipulating the level of expression of the phosphatase CheZ, which deactivates the chemotactic response regulator CheY (Supplementary Fig. 9a). **b** Time-lapse coordinates (colored dots) of an equal mixture of low (red in **a**) and high (cyan in **a**) TB cells traveling in 200 μM aspartate M9 glycerol buffer (see Methods). Scale bar, 0.6 mm. **c** Corresponding density profiles (colors) together with total cell density (black). (Line: mean over n = 34 time points measured at 40 s intervals for one experiment; shading: SD; five replicates are in Supplementary Fig. 9b). **d** Same as in **c**, but for the magenta and green populations in **a**. Two replicates are in Supplementary Fig. 9c. **e** Peak positions as a function of time for the experiment in **b**. **f** The distance between the fluorescence intensity peaks of the two populations traveling together in a single band increases with the difference between the mean TB of the two populations. For each independent experiment, two populations labeled with different fluorescent proteins (mRFP1 or YFP) were induced using different aTc levels to obtain distributions with different mean tumble biases. Dots: average over n = 4 experiments; error bars are SD

maintain traveling cell density over long times[41–43]) to this process.

Previous analysis of bacterial traveling bands assumed that the population consisted of identical cells[10,11,37,44,45] or at most two phenotypes[24,25]. Here we extended these studies by taking into account the continuum of phenotypes that is always present in a population[13,21,22]. One study that considered how two phenotypes might travel together made the theoretical assumption that cells sense only the direction of the gradient, not its magnitude[25]. This assumption causes the peak densities of the two phenotypes in that model to coincide in space, contrary to our experimental observations. In another theoretical study[24], the cells did respond to the gradient magnitude, and the two phenotypes in the traveling solution were spatially separated. Although not discussed in the paper, the phenotype with the higher chemotactic coefficient is in the front in that solution, in agreement with our sorting mechanism. However, in that model,

the range of sensitivity was assumed to extend to vanishing concentrations, as in the original Keller–Segel model[10], which is not biologically realistic[27].

Following depletion of local resources, the spatial self-organizing mechanism described here could enable populations of bacteria to maintain diversity while traveling toward better environments. This diversity increases the probability that a phenotype well-suited to unexpected environments will be available if needed during travel until a destination is reached where growth can replenish the population. As the range of phenotypes allowed within a traveling group depends on the spatial profile of the traveling gradient, this mechanism introduces important feedback between the environment, cellular metabolism, and phenotypic diversity, which together generate spatial patterns of phenotypes according to functional capabilities. The same mechanism might also enable different bacterial species to travel together, thus enabling migration of small ecosystems.

Collective migration of eukaryotes resulting from traveling gradients of attractants generated by consumption or breakdown of an attractant has recently been found to be more important than previously believed because it enables cell migration over much larger distances than migration along externally imposed gradients[46]. Being able to maintain diversity within the traveling group could be important in the context of immunity and cancer.

The types of interactions between collective behavior and phenotypic diversity reported here might also be at play beyond microbiology and cell biology, in contexts where individuals in a group respond to the cumulative effect on the environment of the individuals ahead. Whereas a bird monitors its neighbors to benefit from the collective information acquired by the flock, here individuals monitor the environmental gradient to benefit from the information accumulated in the environment by the band. In both cases, there is a group "memory" in the form of a spatial structure that individuals respond to[47]. In general, both types of memory are probably available and utilized by groups, with their relative importance determined by the specific biology of the organisms.

## Methods

**Mathematical model**. We extended the classic Keller–Segel[10] model to include the effect of phenotypic differences in TB. The key variables in the model are the density $\rho_i(x, t)$ of cells of phenotype $i$ as a function of position $x$ and time $t$, and the concentration of Asp $A(x, t)$. Cells of phenotype $i$ are characterized by their chemotactic coefficient $\chi_i$ and diffusivity $\mu_i$. As Asp consumption depends on oxygen[32], we also model the amount of oxygen dissolved in the media $O(x, t)$. The parameters of the model are in Table 1. The time-dependent evolution reads:

$$\frac{\partial \rho_i}{\partial t} = \mu_i \frac{\partial^2 \rho_i}{\partial x^2} - \frac{\partial}{\partial x}\left(\chi_i \rho_i \frac{\partial f(A)}{\partial x}\right) \tag{1}$$

$$\frac{\partial A}{\partial t} = \mu_A \frac{\partial^2 A}{\partial x^2} - \alpha_A(O)\frac{A}{K_A + A}\sum_i \rho_i \tag{2}$$

$$\frac{\partial O}{\partial t} = \mu_O \frac{\partial^2 O}{\partial x^2} - \alpha_O \frac{O}{K_O + O}\sum_i \rho_i + \kappa(O_{ex} - O) \tag{3}$$

Eq. 1 represents the motion of cells due to diffusion and chemotaxis. $f(A) = M\log\left[\frac{1+A/K_0}{1+A/K_1}\right]$ is the perceived signal which depends on the local Asp concentration $A$, where $K_0 = 3.5\,\mu M$ and $K_1 = 1000\,\mu M$ represent the dissociation constants of Asp for the inactive and active conformations of the Tar receptors, and $M = 6$ is the receptor gain[29,48]. The effective diffusion coefficient and the chemotaxis coefficient are functions of the microscopic parameters of individual cell swimming behavior. We draw on previous work[22,29] to model them as $\mu_i = \frac{v^2}{3}\frac{(1-TB_i)}{(1-\Theta)\lambda_{R,i}+2D_{rot}}$ and $\chi_i = \frac{k_0 TB_i}{TB_0 + TB_i}\mu_i$ (Supplementary Fig. 8a). In these expressions, $v \sim 36\,\mu m\,s^{-1}$ is the cell swimming speed, which when projected in two dimesnions (2D)[49] corresponds to the average speed we measure in our quasi-2D device ($\sim 28\,\mu m\,s^{-1}$); $\Theta = 0.16$ is the directional persistence between successive runs[22]; $D_{rot} = 0.062$ $s^{-1}$ is the rotational diffusion coefficient during runs[21,50]; $\lambda_{R,i} = \omega\sqrt{\frac{TB_i}{1-TB_i}}$ is the rate of switching from the run state to the tumble state; and $\omega = 3.8\,s^{-1}$ is the effective switching frequency[22]. $\mu_i$ and $\chi_i$ are monotonically decreasing functions of the $TB_i$ and $\chi_i/\mu_i \approx 22 = k_0$, apart for the range $0 \le TB_i \lesssim TB_0 = 0.05$, where $\chi(TB_i)$ is increasing as recently observed in experiments where individual cells were tracked in a static gradient of α-methylaspartate (non-metabolizable analog of Asp)[22].

Eq. 2 represents the change in the concentration of Asp due to diffusion, with diffusivity $\mu_A$, and to consumption, with half-max constant $K_A$ and maximum rate $\alpha_A(O) = \alpha_{A0}\left(1 - g_A + g_A\frac{O}{O_{ex}}\right)$. Here, $\alpha_{A0}$ is the base consumption rate, $O_{ex}$ is the external oxygen concentration, and $g_A$ is the fractional reduction of Asp consumption rate at zero oxygen. Eq. 3 describes the time-dependent evolution of oxygen, with diffusion coefficient $\mu_O$, maximum consumption rate $\alpha_O$, half-max constant $K_O$, and supply of oxygen through the polydimethylsiloxane (PDMS) with the mass transfer rate $\kappa$.

Eqs. (1–3) were integrated in MATLAB using second-order centered differences for the spatial derivatives (mesh size 20 μm) and an explicit fourth-

order Runge–Kutta routine for temporal integration (time step 0.08 s). We used no-flux boundary conditions. The initial condition was $\rho_i(x, 0) = \rho_0 \cdot P(TB_i)e^{-(x/x_0)^2}$ for $0 < x < 1.6$ mm and $\rho_i(x,0) = 0$ for $x > 1.6$ mm. Here, $\rho_0$ is the initial cell density scale, determined from the total initial cell number $\sim 2 \times 10^5$; $x_0 = 0.8$ mm; and $P(TB_i)$ was obtained from experimental measurements (Fig. 2e black).

Assuming near-constant wave speed, we rewrite Eq. 2 in the moving coordinate $z = x - ct$ and integrate from $-\infty$ to $+\infty$ to obtain

$$\langle \alpha_A \rangle = \frac{cA_0}{N/a}. \tag{4}$$

Here, $\alpha_A$ is the average attractant consumption rate, $N$ is the number of cells in the band, and $a$ is the cross-sectional area of the channel. In the absence of oxygen-dependent consumption of Asp, the average consumption rate in the band would be constant across experimental conditions[10]. However, oxygen-dependent consumption makes the average consumption rate decreases with increasing cell density in the band, which is correlated with the number of cells in the band (Fig. 2c). As shown in Supplementary Fig. 7h, the value of $\langle \alpha_A \rangle$ calculated from experimental data using Eq. 4 decreases as the number of cells in the band increases, which is captured in simulations.

To derive the result of spatial sorting analytically from this model, we first rewrite Eq. 1 in the moving coordinate $z = x - t$:

$$-c\frac{\partial \rho_i}{\partial z} = \mu_i \frac{\partial^2 \rho_i}{\partial z^2} - \frac{\partial}{\partial z}\left(\chi_i \rho_i \frac{\partial f(A)}{\partial z}\right), \tag{5}$$

Noting that for each phenotype $i$ to be traveling with the group, its density profile must have a peak. Around the density peak $z_i$ we must have $\frac{d\rho_i}{dz}\big|_{z=z_i} = 0$ and $\frac{d^2\rho_i}{dz^2}\big|_{z=z_i} < 0$.

Rewriting Eq. 5 as

$$\mu_i \frac{\partial^2 \rho_i}{\partial z^2} = -c\frac{\partial \rho_i}{\partial z} + \chi_i \frac{\partial \rho_i}{\partial z}\frac{\partial f(A)}{\partial z} + \chi_i \frac{\partial^2 f(A)}{\partial z^2}, \tag{6}$$

we then have at the density peaks $z_i$:

$$\mu_i \frac{d^2\rho_i}{dz^2}\bigg|_{z=z_i} = \chi_i \rho_i \frac{d^2 f}{dz^2}\bigg|_{z=z_i} < 0 \tag{7}$$

Eq. 7 indicates that in the front of the band the perceived gradient must be shallow and become progressively steeper as $z$ decreases toward the back (Fig. 3c black). Integrating Eq. 5 and using $\frac{d\rho_i}{dz}\big|_{z=z_i} = 0$, we obtain

$$c = \chi_i \frac{df}{dz}\bigg|_{z=z_i} \tag{8}$$

Eqs. 7, 8 together show that the cell density peaks $z_i$ of each phenotype $i$ are monotonically ordered according to their chemotactic coefficients $\chi_i$.

Examining the effect of oxygen analytically, at the back of the wave the Asp concentration is small so that the chemotactic drift there is $\chi_i \frac{\partial f(A)}{\partial z} \approx \chi_i \frac{M}{K_0}\frac{dA}{dz}$. Rewriting Eq. 2 in the moving coordinate, assuming the diffusion term is negligible[10], and assuming $A \gg K_A$ gives an expression for $dA/dz$. From this, the drift becomes

$$\chi_i \frac{\partial f(A)}{\partial z} \approx \frac{\chi_i M}{K_0 c}\alpha_A(O)\sum_i \rho_i. \tag{9}$$

At the back of the band there are fewer cells and therefore more oxygen than in the middle of the band. Thus, when the Asp consumption rate depends on oxygen, $\alpha_A(O)$ becomes larger at the back than the mean over the band, $\langle \alpha_A \rangle$. As a consequence, the drift at the back is higher than in the case without oxygen dependence, slowing down the decay of the band. We verified this by running simulations with and without oxygen dependence (Supplementary Fig. 8b). In the simulations without oxygen, we set $\langle \alpha_A \rangle$ equal to a constant value, corresponding to the average consumption rate in the band $\alpha_A$ in the simulation with oxygen dependence. This was intended to make the wave speeds in the two simulations similar, eliminating the effect of different wave speeds on cell leakage rates.

**Strains, growth conditions, and sample preparation**. E. coli RP437 was used as the wild type strain for chemotaxis in this study. Cells were grown in M9 glycerol medium: M9 salts (6.78 g L$^{-1}$ Na$_2$HPO$_4$, 3.0 g L$^{-1}$ KH$_2$PO$_4$, 0.5 g L$^{-1}$ NaCl, 1.0 g L$^{-1}$ NH$_4$Cl), supplemented with 4 mL L$^{-1}$ glycerol, 0.1 % casamino acids, 1.0 mM magnesium sulfate, and 0.05% w/v polyvinylpyrrolidone-40 at 30 °C. Appropriate antibiotics were supplemented (ampicillin 100 μg mL$^{-1}$, kanamycin 50 μg mL$^{-1}$, and chloramphenicol 25 μg mL$^{-1}$) when necessary to maintain plasmids.

For Fig. 1, cells were collected at mid-exponential phase (approximately an OD600 of 0.3) and washed twice with fresh M9 glycerol medium, then resuspended

**Table 1 Model parameters**

| Symbol | Definition and value | Reference |
|---|---|---|
| $M$ | The receptor gain for aspartate, $M = 6$ | Ref. [29] |
| $K_O$ | The dissociation constant to aspartate for the inactive conformation of the Tar receptor, $K_O = 3.5\ \mu M$ | Ref. [48] |
| $K_1$ | The dissociation constant to aspartate for the active conformation of the Tar receptor, $K_1 = 1000\ \mu M$ | Ref. [48] |
| $\mu_A$ | The diffusion coefficient of aspartate molecules, $\mu_A = 500\ \mu m^2\ s^{-1}$ | Ref. [55] |
| $K_A$ | The aspartate concentration at half-max of its consumption, $K_A = 0.5\ \mu M$ | Ref. [56] |
| $\alpha_{AO}$ | The maximum aspartate consumption rate, $\alpha_{AO} = 9.3 \times 10^{-12}\ \mu mol^{-1}\ cell^{-1}$ | Supplementary Fig. 7g |
| $O_{ex}$ | The external oxygen level, $O_{ex} = 250\ \mu M$ | Ref. [57] |
| $g_A$ | The basal ratio of relative consumption rate at zero oxygen, $g_A = 0.27$ | Supplementary Fig. 7g |
| $\mu_O$ | The diffusion coefficient of dissolved oxygen, $\mu_O = 2500\ \mu m^2\ s^{-1}$ | Ref. [55] |
| $\alpha_O$ | The maximum oxygen consumption rate, $\alpha_O = 7 \times 10^{-11}\ \mu mol\ min^{-1}\ cell^{-1}$ | Supplementary Fig. 7g |
| $K_O$ | The dissolved oxygen concentration at half-max its consumption, $K_O = 1\ \mu M$ | Supplementary Fig. 7g |
| $\kappa$ | The oxygen transfer rate through ~ 0.5 cm of PDMS, $\kappa = 0.02\ s^{-1}$ | Ref. [58] |
| $\rho_0$ | Initial cell density, $\rho_0 \sim 2.87 \times 10^{10}$ cells per mL | This study |
| $x_0$ | Length scale of initial cell density profile, $x_0 = 0.8$ mm | This study |
| $v$ | Cell swimming speed, $v \sim 36\ \mu m\ s^{-1}$ | This study |
| $\Theta$ | Directional persistence $\Theta = 0.16$ | Ref. [22] |
| $D_{rot}$ | Rotational diffusion coefficient during runs, $D_{rot} = 0.062\ s^{-1}$ | Ref. [50] |
| $k_0$ | Parameter in the expression for $\chi_i/\mu_i$, $k_0 = 22$ | Ref. [22] |
| $TB_0$ | Parameter in the expression for $\chi_i/\mu_i$, $TB_0 = 0.05$ | Ref. [22] |

in fresh M9 glycerol medium to concentrate cell density at an OD600 of 0.7. These cells were then gently loaded into the microfluidics chamber, which was maintained at 30 °C throughout the experiment.

To generate a single traveling band, experiments were conducted in M9 glycerol buffer: motility buffer (M9 salts, 0.01 mM methionine, 0.1 mM EDTA, 0.05% w/v polyvinylpyrrolidone-40) supplemented with 4 mL L$^{-1}$ glycerol and the indicated amount of Asp. This buffer was used to wash and resuspend cells instead of the complex growth medium (M9 glycerol medium) mentioned above. The same M9 salts were used in the M9 glycerol medium and M9 glycerol buffer to minimize osmolality changes. RP437 is auxotroph for leucine, histidine, methionine, and threonine, and therefore does not grow in M9 glycerol buffer.

To control the TB distribution, we used a $\Delta cheZ$ strain derivative of RP437 containing a chromosomally-integrated copy of the phosphatase CheZ under the control of the inducible promoter and *tetR* (gift from Dr. Chenli Liu). CheZ dephosphorylates the response regulator CheY, which, when phosphorylated, induces the motors to switch and the cells to tumble. Thus, low CheZ results in higher CheY-P and more tumbling, and vice versa. aTc was added overnight in the culture when indicated to release the repressor TetR from the *cheZ* promoter region, inducing the expression of CheZ in this strain. To color-code strains, pBca1020-r0040 carrying mRFP1 (obtained from BioBrick), pLambda driving mRFP1 (gift from Dr. Chenli Liu) and plasmids carrying YFP[51] under constitutive promoter were transformed into RP437 and into the inducible CheZ strain by electroporation.

**Microfluidic device design and fabrication**. Microfluidic devices were constructed from the biocompatible and oxygen-permeable silicone polymer PDMS on cover glass following standard soft lithography protocols for two-layer devices[52]. The master molds for the device consisted of two silicon wafers with features created using ultraviolet (UV) photoresist lithography. The bottom wafer had two main parts: a large chamber created using SU-8-negative resist (thickness: 10 μm, SU8 3010, Microchem) and a long channel together with two inlet/outlet channels designed to be opened and closed using pressure actuated valves. A second coat of SPR-positive resist (thickness: 14 μm, SPR 220-7.0, MEGAPOSIT) on the same wafer was used to create a rounded channel profile that can collapse fully if depressed from above (Supplementary Fig. 1a). The second, top wafer contained features for the control channels that close the collapsible features in the bottom wafer. The top wafer was created using SU-8 negative resist (thickness: 10 μm, SU8 3010, Microchem). The resists were then cured using UV light exposure through photomasks designed in CAD software and printed by CAD/Art Services, Inc. (Bandon, Oregon), again following photoresist manufacturer specifications. Subsequently, wafers were baked and the uncured photoresist was dissolved. After curing the SPR coat, the features were baked further to produce a rounded profile. After both wafers were complete, a protective coat of silane was applied by vapor deposition.

To cast and manufacture the two-layer device, the top wafer was coated with a 5 mm-thick layer of degassed 10:1 PDMS-to-curing agent ratio (Sylgard 184, Dow Corning). For the bottom layer, a 20:1 mixture was prepared and spin coated to create a 100 μm-thick layer. The two layers were partially baked for 45 min at 70 °C. The top layer was then cut and separated from the wafer, holes were punched from the feature side using a sharpened 20-gauge blunt-tip needle to make external connections to the control valve lines, then aligned and laminated onto the bottom layer. The stacked layers were baked together for 1.5 h at 70 °C and allowed to cool.

The laminated layers were then cut out and the remaining ports were punched to make external connections with the channels. To reduce the evaporation of the microfluidic device, the PDMS device was soaked overnight in Millipore-filtered water at 50 °C.

The assembled PDMS devices were bonded to 24 × 50 mm glass coverslips (#1.5). The PDMS was cleaned with transparent adhesive tape (Magic Tape, Scotch) followed by rinsing with (in order) isopropanol, methanol, and Millipore-filtered water, air-drying between each rinse. The glass was rinsed the same way, with acetone, isopropanol, methanol, and Millipore-filtered water. The PDMS was tape-cleaned an additional time, and then the two pieces were placed in a plasma bonding oven (Harrick Plasma) under vacuum, gently laminated, and then baked on an 80 °C hotplate for 15 min to establish a covalent bond. Devices were stored at room temperature and used within 24 h.

**Band formation and imaging**. Washed cells were gently loaded into the device which was then centrifuged for 20 min at 700 g in a 30 °C environmental room to concentrate cells at the end of the chamber (Supplementary Fig. 1b). After spinning, the microfluidic device was placed on an inverted microscope (Nikon Eclipse Ti-U) equipped with a custom environmental chamber (50% humidity and 30 °C). A custom MATLAB script was used to control the microscope and its automated stage (Prior) via the MicroManager interface[53]. Time-lapse images (phase-contrast and fluorescence: mRFP1 or YFP) of the migrating cells were acquired using a Hamamatsu ORCA-Flash4.0 V2 camera (2,048 × 2,048 array of 6.5 × 6.5 μm pixels), a ×10 phase contrast objective (Nikon CFI Plan Fluor, N.A. 0.30, W.D. 16.0 mm) and a LED illuminator (Lumencor SOLA light engine, Beaverton, OR) through the mCherry block (Chroma 49008, Ex: ET560/× 40, Em: ET630/75 m) or EYFP block (Chroma 49003; Ex: ET500/× 20, Em: ET535/30 m). Once the band formed, starting at the origin (closed end of the channel), the motorized stage moved along the channel and paused every ~ 1.3 mm (the width of one frame with a small overlap < 0.1 mm between consecutive positions) to take images in phase contrast and fluorescence (exposure time 122 ms for both channels). After reaching the observation chamber, acquisition started over at the origin every 40 s (every 60 s for Fig. 1c).

In Fig. 1, all cells were expressing mRFP1. In Fig. 2, unlabeled and fluorescently labeled cells were separately grown to mid-exponential phase (OD600 of about 0.3). For the experiments with 50, 100, and 200 μM Asp, the fluorescently labeled cells were diluted with unlabeled cells at the following ratios 1:20, 1:50, 1:100. The mixed cells were then washed and resuspended to the same predetermined density (OD600 of 0.7). The mixed populations were loaded into the microfluidic device and imaged as described above. In Fig. 4b-e, a similar procedure was followed to prepare the samples for different induction conditions. A 1:1 mixture of high and low induction cultures was mixed and loaded in the device.

In Fig.4f, a 1:1 mixture of high and low aTc induction (ranging from 1 to 10 ng mL$^{-1}$) cultures with all cells fluorescently labeled by mRFP1 or YFP was mixed and loaded in the device. The distances between the peaks of the density profiles of the two populations were calculated by measuring the distance between the peaks in the two fluorescent intensity profiles. The mean TB of each population were measured by loading a sample of the population on a cover slip and tracking individual cells as previously described[21].

Once a band arrived in the perfused chamber, the gate near the chamber was closed (using 10 psi pressure) to capture the band (Supplementary Fig. 1b). To capture the second band in a separate experiment, the gate remained open until the

first band passed, then closed. The first band was immediately flushed away by flowing M9 glycerol medium at 3 psi for 5 ~ 20 s through the chamber until all cells were flushed from the chamber. The gate was then reopened to let the second band migrate into the chamber. Once a band was captured in the perfused chamber, several pulses of fresh medium/buffer at 3 psi were flown in to reduce cell density and homogenize the environment in the chamber. Cells were left to adapt in the perfused chamber for a few minutes.

**TB detection**. Once cell density became relatively homogeneous, the swimming trajectories of individual cells were recorded for 2 min at 8 fps in order to extract TB. We verified that perfusion of the observation chamber did not affect the distribution of TBs (Supplementary Fig. 3). Each TB distribution was generated by acquiring four movies, tracking the individual cells, and determining their TB as described before[21,22]. The resulting number of sample trajectories longer than 10 s (shorter tracks were discarded, because they provide poor estimate of TB[21]) was limited by the number of independent movies one can acquire in the PDMS chamber, the size of which corresponds to four field of views, and the density of cells, which must be kept low for tracking to be possible. This procedure resulted in a minimum of 2823 and 3209 trajectories per distribution giving an error of at most 1.15% and 1.00% in the determination of the cumulative distribution function (CDF) estimated by bootstrapping for Figs. 1 and 2, respectively.

**Determination of the number of cells in the band**. Image analysis was conducted in MATLAB. We detected the position of the centroid of each fluorescent cell using the MATLAB function bwconncomp. Figures 2a, 4b, and Supplementary Figs. 6d, e report these coordinates. The number of labeled cells was multiplied with the dilution ratio to obtain the total number of cells in the band.

Cell density profiles in Fig. 2d, 4c d, and Supplementary Figs. 9b, c were measured as follows: cell density profiles were extracted for each time point within one experiment and aligned before averaging. The cell density profile at a given time point was calculated by dividing the number of cells by the volume in one spatial bin (~ 120 μm) along the observation channel. To reduce alignment error before averaging, the position of the peak density in each profile was identified by first smoothing the profile with a moving filter with a 5-bin span (MATLAB function smooth) and then identifying the position of the peak. To avoid boundary effects, only the profiles with peak position located between 3 and 8 mm from the origin were used to calculate the average density profile. The mean and SD shown in the figures were calculated with raw cell density profiles (not smoothed).

**Measurement of amino acids by HPLC**. When consuming Asp, E. coli cells secrete other amino acids, which could affect group migration[54]. We used HPLC to analyze the amino acids secreted by the cells when they are suspended in M9 glycerol buffer supplemented with Asp. RP437 cells were grown in 200 mL M9 glycerol medium up to mid-exponential phase and washed twice with M9 glycerol buffer supplemented with 500 μM Asp. Cells were then resuspended in 5 mL of the same defined buffer at an OD600 of 1 and placed in a 200 mL flask. The flask was shaken at 200 r.p.m. to maximize aeration at 30 °C. Every 15 min, a 500 μl of culture was sampled, filtered using 0.2 μm filter (Acrodisc 13 mm Syringe Filter with 0.2 μm HT Tuffryn Membrane, Pall Corporation), and analyzed by HPLC via pre-column derivatization method. The resulting derivatives were separated by phase chromatography using a Dionex Ultimate 3000 HPLC, with a coupled DAD-3000RS diode array detector (Dionex) and FLD detector (Dionex) using an ACE C18 column (3 μm, 3 × 150 mm). Amino acid standard (AAS18 Sigma) was used as reference.

Upon uptake of Asp, cells secreted small amounts of glutamate (Glu), asparagine (Asn), and homoserine (HS), which are attractants (Supplementary Fig. 6c). To quantify the relative contribution of each amino acid to the chemotactic response in our experiment the measured concentration of each amino acid was plotted in units of the corresponding EC50 of the dose response of the chemotaxis system. The EC50 of the dose response of the chemotaxis system for each amino acid has been quantified by the Sourjik lab using in vivo Förster resonance energy transfer measurements in RP437, the strain used in this study. The EC50 values of RP437 E. coli are 0.3 μM for Asp, 50 μM for Glu, 30 μM for Asn, and 3 mM for HS[48]. Supplementary Fig. 6c reveals that the response to Asp dominates by almost two orders of magnitude over the responses to the other amino acids. It also shows that of the three secreted amino acids, Glu is the one that has the second largest effect, albeit still much smaller than the response to Asp. Finally, we also checked that chemotaxis towards oxygen does not play a significant role either. Mutant strains lacking the oxygen receptor aer or both aer and tsr form bands under the same condition as in Fig. 2a, indicating that aerotaxis is not essential and Tar response to Asp is sufficient for the band to travel (Supplementary Fig. 6d, e).

**Measurement of oxygen in the center of the traveling band**. Ruthenium complexes are toxic to E. coli; hence, the need to encapsulate them in phospholipid micelles. This was achieved using the same protocol as in ref. [33]. Fluorescence of the ruthenium complex is quenched by oxygen binding, so higher fluorescence corresponds to lower oxygen concentration. It is noteworthy that the high density of cells in the band could exclude the dye; however, this should decrease

fluorescence intensity, the opposite effect of decreased oxygen. Experiments were performed in 100 μm-deep and 14 μm-deep devices. The 100 μm-deep devices were straight channels, 39 mm long and 600 μm wide. The 14 μm-deep devices were the same as those used in the other experiments in this paper. However, as the control gates interfere with the fluorescence signal, the top layer of PDMS was fabricated without a mold for the gates. In both cases, a 1:500 dilution of concentrated micelles was added to bacteria prepared as described above, just before loading into the microfluidic device. This dilution was intended to avoid sequestration of oxygen from the cells. Two control experiments were performed in the 14 μm device, one with dye and no cells, and one with cells and no dye. Imaging and data analysis of all three types of experiments were performed in the same way.

Imaging equipment were the same as described above with a few exceptions. The excitation filter from an ECFP block (Chroma 31044v2, D436/× 20), the emission filter from an mCherry block (Chroma 49008, ET630/75 m), and the dichroic mirror from the mCherry block (Chroma 49008, T585lpxr) were used to image the dye due to its large Stokes shift.

For the 100-μm device, the × 10 objective described above was used for imaging. Images were taken in 2 min intervals. For the 14-μm device, imaging was performed using a × 40 oil objective (Nikon CFI Plan Fluor, NA 1.3, W.D. 0.24 mm). Images were taken in 3.5 min intervals.

To analyze the fluorescence images, fluorescence intensity was first averaged over the width of the channel that was visible in the image, $I(x, t)$. At each location $x$, the passing wave appeared as a brief peak in intensity during the time course. To separate this from slow variations in signal due to photobleaching and possible global changes in oxygen concentration, we smoothed the time course of fluorescence intensity at each position, $I_x(t)$, using MATLAB's smooth function (smoothing method: lowess; window size: 4, corresponding to a time window of 14 min) to produce the slowly varying background signal, $\hat{I}_x(t)$. To extract the fast-passing wave, we divided $I_x(t)$ element wise by $\hat{I}_x(t)$ for each position $x$. As a result, the slowly changing background was normalized to 1, while faster changes in signal were different from 1. This also eliminated differences in illumination over space. Finally, the noisy normalized profiles produced by this analysis were median filtered over space using MATLAB's medfilt1 (window size: 10).

**Oxygen consumption rate in batch cultures**. RP437 cells were grown in 200 mL M9 glycerol medium to mid-exponential phase and washed twice with M9 glycerol buffer supplemented with 200 μM Asp. Cells were then resuspended in 50 mL of the same defined buffer to an OD600 of 0.5. The sample was placed in a beaker at 30 °C. The surface of the buffer was sealed by overlaying mineral oil. The level of dissolved oxygen was measured with a portable dissolved oxygen meter (Melwaukee MW600) every 30 s. The cell sample was continuously stirred at 300 r.p.m. with a magnetic stirrer. The consumption rate per cell per minute was obtained by dividing the reduction in oxygen by the number of cells and the sampling interval time (Supplementary Fig. 7g).

**Asp consumption rate in batch cultures**. RP437 cells were grown in 200 mL M9 glycerol medium up to mid-exponential phase and washed twice with M9 glycerol buffer supplemented with 500 μM Asp. Cells were resuspended in 3 mL of the same defined buffer to make cell density at an OD600 of 1. Two samples were prepared in two test tubes. One tube was shaken at 200 r.p.m. to maximize aeration, whereas the other tube was left on the bench with mineral oil overlaid on the liquid surface to avoid supply of oxygen from air. These two tubes were incubated at 30 °C and sampled every 10 min. The amount of Asp in the collected sample was then measured by using Aspartate Assay Kit (Abcam). The consumption rate per cell per minute was obtained by dividing the reduction of Asp in the sample by the number of cells and the sampling interval time (Supplementary Fig. 7g).

**Statistical analysis and experimental reproducibility**. No statistical methods were used to predetermine sample size. Standard error in the CDF of the TB in each replicate experiment was determined by bootstrapping (1000 bootstrap samples). One-sided unpaired two-sample Student's t-test assuming unequal variances was used for comparison between two groups in Fig. 2b, c. P-values < 0.05 were considered statistically significant and marked with asterisks. For t-test, t-values and degrees of freedom are provided in the figure legends. The error bars are defined in each figure caption and are standard deviation except in Fig. 2f. Data presented in the main figures were drawn from at least three independent replicates, with the exception of Fig. 4d ($n = 2$). The number of replicates is mentioned in the caption of each figure.

**Code availability**. To analyze the swimming behavior of E. coli cells we used custom MATLAB code as reported in refs. [21,22]. The code to simulate the mathematical model is described above and available from the corresponding author upon request.

**Data availability**. Data for each figure is provided as a MATLAB .fig file from which the data points can be extracted https://doi.org/10.6084/m9. figshare.6207371.

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

## Acknowledgements

We thank H. Salman for sharing the protocol to synthesize the Ru-micelles; Y. Dufour for help with the microfluidics; F. Isaacs for sharing BioBrick plasmids; C. Liu, S. Parkinson, and H. Salman for sharing *E. coli* strains and plasmids; N. Clay, B. Barco, and W. Chezem for help with HPLC experiments; D. Clark and J. Wang for help with statistics;, and A. Waite, J. Howard, C. Brannon, A. Greene, and P. Turner for comments on the manuscript. This study was supported by the National Institutes of Health grant 1R01GM106189 and the Allen Distinguished Investigator Program (grant 11562) through The Paul G. Allen Frontiers Group. C.H. acknowledges support through the National Natural Science Foundation of China (31570095, 11504399).

## Author contributions

X.F. and S.K. contributed equally to this work. H.M. and J.L. contributed equally to this work. T.E., X.F., and S.K. designed the research. X.F., S.K., H.M., and C.H. performed the experiments. X.F., S.K., J.L., H.M., and T.E. performed the data analysis. X.F., J.L., H.M., and T.E. performed the mathematical modeling. D.C.V. and S.W.Z. contributed to mathematical modeling. T.E., X.F., S.K., H.M., and J.L. wrote the manuscript. All authors discussed the manuscript.
