## [Peer Review File · Nature Communications]

Reviewers' comments:

Reviewer #1 (Remarks to the Author):

This manuscript by Fu and colleagues uses bacterial chemotaxis as a model system to ask a general question: how can organisms migrate as a group despite differences between individuals in the ability to move? This manuscript builds on previous work by this lab and others that recorded marked differences between individual bacterial cells in the ability to chemotax in gradients of attractants. At least part of these differences between individuals are based on variation in the 'tumbling bias' – the frequency at which cells interrupt runs by tumbles. Tumbling bias directly translates to a cell's ability to swim up a gradient of an attractant. One might thus expect that cells with different tumbling biases would become separated from each other during chemotaxis in a gradient. However, and intriguingly, this is not the case; despite the variability in chemotaxis performance, these cells migrate as a group. The paper describes how this works, and thereby provides a new and fundamental insight into the link between individuality and collective behavior.

The main discovery put forward by the authors is that the individuals with the strongest chemotactic performance change the local environment (the gradient of the chemoattractant) in a way that allows individuals with lower performance to keep up with the rest. A number of quantitative single-cell experiments support the main aspects of this idea: the authors construct different mutants that vary in their chemotactic performance and show that these mutants can migrate together; and they show that the spatial organization of the mutants corresponds to their prediction, with the types with the strongest performance traveling at the front and the types with weaker performance further back (but still as part of the same group, defined as a moving unimodal distribution of bacterial density along the gradient.)

This manuscript addresses a fundamental question ('how does collective behavior emerge in heterogeneous groups'), it puts forward an intriguing new concept, and supports it with conclusive experimental data. I thus think that the paper will make a seminal contribution, and be of interest to scientists broadly interested in collective behavior, complex biological systems, and to scientists interested in the microbial sciences.

I have a few of suggestions and comments.

3th paragraph discussion ("Because the range of phenotypes...")

- The phrasing "spatially allocates labor" can be misinterpreted to suggest that cells in different parts of the wave perform different tasks (as in division of labor), however the authors showed that all cells perform the same task, but perform this task with different proficiency. I suggest to rephrase this sentence.

- "...suggest a role for non-genetic diversity in generating functional structure within cellular communities." This phrasing suggests that by sorting cells with different chemotactic performance in the migration wave new functionality is achieved at the group level. It is not clear to me whether this is the right formulation. Without phenotypic heterogeneity cells can form migrating bands. Phenotypic heterogeneity could potentially prevent these bands from forming, but the sorting described by the authors provides a solution to this problem. I think it is thus fair to say that sorting allows functionality at the group level in spite of the potential debilitating effects of heterogeneity, but I would not call this new functionality. I would suggest to rephrase this sentence.

- "In the presence of genetic variation ...". It is not clear to me what exactly the authors mean with this statement and how it is related to the findings shown in the manuscript. I suggest to remove or clarify this statement.

Fig 2

"...the cell density profile became narrower (Fig. 2d)", page 5

The large difference in the maximum cell density between conditions makes it hard to compare the distributions in Fig. 2d to validate this statement. It would be helpful to have a quantification of the width of the distribution or to show normalized graphs of the density profile.

"Here, we assume the self-attractant to be glutamate..." Page 14 SI

Why did the authors choose glutamate? Is this a conservative assumption? Or can the authors give a rough estimate of how sensitive their conclusions are to the choice of the K_0 and K_1 values? I don't think that a full sensitivity analysis is needed, but I think it would be useful for readers to get an idea of how strongly the conclusions depend on this assumption.

"...slow decay due to cells falling off at the back of the band", page 5

In several places the authors mention that densities in the band decrease due to cells falling off the back of the wave. Did the authors also consider the possibility that some cells escape at the front (i.e. run ahead of the wave)? Can the relative importance of these two possibilities be estimated, or is there an a-priori reason to assume that most loss is at the back?

Minor comments:

Fig 2b,c:

The difference between the diamond and circle symbols is not explained in the caption.

Fig 6e,f,

To facilitate comparison of data and model it would be very helpful to add the experimental data to these panels.

Fig 8b,c:

The caption does not include a description of the meaning of the colors of the lines

Reviewer #2 (Remarks to the Author):

This article investigates the impact of diversity in collective chemotaxis of bacteria. It combines experimental observations and theoretical analysis.

This is an interesting idea that has been previously explored (see in particular Emako et al. PLoS Comp Biol (2016)). The experiments are very carefully performed and the collected data appears to be of high quality. For the sake of clarity, I suggest to move the set of experiments performed in presence of casamino acids that remains very qualitative, to the Supplementary Material. The core of the paper deals with the more controlled experiments performed in presence of aspartate only and their interpretation by the proposed theoretical model.

On the experimental side, more details on the microfluidic setup should be given. The fabrication of the device could be made more explicit with a schematic. If I understand correctly, the depth of the PDMS channel is only 10 μm (smaller than a run length). How can the authors then rule out the influence of the surfaces that has been evidenced in many studies? This small depth may be related to the large pulse velocities (larger than 10 $\mu\text{m}/\text{s}$). Surprisingly, the band speeds and densities are measured with mixtures of labeled and unlabeled cells (p5), although the reason for that is not clear. This is problematic as the emphasis of the paper is put on having a pure clonal population. Fig. 2C shows that the number of fluorescent cells increases with [Asp]. It is true of the total number? In the same proportion?

Since the experiments basically confirm previously published results, the true originality of the paper is the theoretical model. It is based on a modified version of the Keller Segel model that includes the concentration of oxygen in the medium and a dependency of aspartate consumption with oxygen concentration. This is a non-trivial component as experiments are performed in PDMS microchannels, a material very permeable to Oxygen, in very shallow channels. As mentioned by the authors themselves, a more common way to write these KS equations is to involve an

attractant secreted by the bacteria. Some arguments in favor of the importance of oxygen consumption vs attractant secretion are given in the Supplementary Notes. However, they remain very qualitative. Besides, Douarche et al. PRL (2009) have shown that the diffusion coefficient of the bacteria is a highly non-linear function of oxygen concentration. Shouldn't this effect or simply aerotaxis be then included in the equations? Since the model is the truly original part of the paper, it appears absolutely critical to directly measure the local oxygen concentration in the channel (Douarche et al. PRL 2009) to validate the different hypotheses in a more direct way than the present indirect experiments listed in Supp Fig 7.

As the article stands now, the comparison between theory and experiments remains too qualitative and the proposed model describes the experiments no worse or better than other models that include chemoattractant secretion. The dependence of the diffusion coefficient with bacterial density (Wu et al. PRL 2000) and Saragosti PNAS (2011)) should also be included. It is surprising that no figure compares directly and quantitatively experiments and simulations. In the table recapitulating the results, many parameters are not explicitly given. Moreover, the 4 strains engineered to have different TBs are not characterized independently. Why isn't the behavior of each of these strains studied separately (pulse velocities, diffusion coefficients, chemotaxis coefficients), and compared to theory? How does the peak velocity of a given strain alone compares with the one it acquires when mixed with another one? Looking closely at Fig. 4b, it looks like the two bands travel at slightly different velocities and therefore may end up separated if one waits sufficiently long. In other words, the distance between the peaks in Fig. 4c is itself an increasing function of time. It would be useful to add the kymographs of these experiments at longer times. At what time point were the Fig. 4c, 4E or 4F measured? What did guide the choice of this particular time? In a similar note, Fig 3b and 4f don't seem to fit: the distance between peaks is larger in the simulations compared to the experiments. Again, a direct comparison would be welcome.

Finally, would it be possible to use gated channels as described in the first part to extract the tumble bias of each population while in the wave, or to obtain this information by analyzing the trajectories of the bacteria in situ (within the wave)?

Reviewer #3 (Remarks to the Author):

In this work, the authors study a clonal population of motile *E. coli* cells that undergo collective migration in a channel filled with nutrients. This collective behavior is fueled by the fact that the bacteria consume the nutrient generating a chemical gradient of nutrient concentration to which they respond. Using a nicely conceived microfluidics device, the authors can obtain measurements of a non-homogeneity of the tumbling bias of the bacteria at different positions within the traveling band. Based on their observations the authors propose a model to justify how bacterial populations with such a heterogeneous phenotype are able to migrate collectively.

The mechanism proposed in this work for keeping bacteria of different tumbling bias swimming together in the same band instead of falling apart is original and very interesting. Indeed, this procedure enables bacteria to achieve this without a chemotactic substance produced by the bacteria themselves and to which they respond giving them a natural way of keeping together which was an essential component of the mechanisms described in many previous works).

Even when there is a self-attraction term, traveling together is not granted and depends on more intricate phenomena : it had even been shown that in case of a traveling speed phenotype difference, bacteria could travel as a single pulse or split into two separate pulses depending on their relative abundance in the population (see reference [9] of the supp. material). This gives a great value to having been able, in the present work, to propose a completely different model that produces collective behavior without the need of a self-attractant and which is coherent with the experimental observations obtained concerning the spatially heterogeneous tumbling bias phenotype inside the traveling band.

However, the experimental evidence for certain parts of this model, in particular the role of oxygen consumption, is rather circumstantial and I think that the value of this paper would be greatly improved if the authors could produce direct evidence that the actual oxygen distribution in the channel agrees with that predicted by their model. There exist fluorescent markers for oxygen described in the literature (see, for instance, C. Douarche et al. PRL 102, 198101 (2009)). Having the same type of information concerning space and time dependence of aspartate concentration would also be nice to check equation 2 of the model (in particular how the aspartate consumption depends on oxygen), but this should be harder.

Another important point that is not very clear in the paper and would deserve being clarified is if the traveling bands described in this paper are a transient phenomenon or if they are intrinsically permanent in ideal conditions (with or without considering proliferation for longer time scales). In fact, on the one hand the loss of mass in the peaks seems to not be limited to the grey areas (see fig. 3d) and on the other the propagation speed seems to decrease (visual impression from fig. 2a and supp. fig. 7). In general, we have nearly exclusively data about distribution profiles corresponding to snapshots (for which the time is sometimes not given) and very little about how things change in time. It would be important to give more information on the evolution in general. For instance how far are we from the idealized situation of a peak of constant mass traveling at constant speed? (which is an important issue for the choice of appropriate mathematical models). If I understood properly, in supplementary figure 6f the authors just indicate that their simulations predict an important density decrease if there is no oxygen dependence on aspartate consumption. What about the experimental results?

One more issue that worries me concerning the data obtained using fluorescently labeled cells to deduce general properties of unlabeled populations (and which is not specific to this work) is that the results presented in supp. reference [9] showed that two e.coli cell populations, where the only difference was the fluorescent label, had very different physical behaviors (in particular different pulse speeds in a similar channel migration experiment). Even if the results obtained seem to be coherent with the assumption that the fluorescent labeling has little or no influence, it would be important for the authors to check carefully what are the consequences of the labeling in their experimental setting.

As a general remark, I find it unfortunate that the mathematical contents are nearly all "hidden" in the supplementary material. In this paper, the mathematical model is crucial for giving impact to the experimental observations - the observation of the spatial structuring of the band in tumbling bias phenotype without the plausible explanation given by the mathematical modeling would have considerably less value. Considering its importance in this study, the mathematical material should be included in the main paper, at least in the methods section (as the same group did in supp. ref. [6]). And some more important aspects that single out the contribution of this work relative to previous mathematical studies, deserve to have a place in the main text, even if they are not always appealing for biological readers. This is also important for them to progressively learn how to appreciate mathematical modeling contributions.

To end, just a couple of short remarks:

- I was puzzled by the precision of the expressions in equations 4 and 5 of the supp material when we see the degree of variability of the experimental results (like the one we see in figure 4f for the mean tumble bias difference of 0.3).
- Doesn't having a 10 micrometer thickness, which is not big compared to the mean run distance, pose a problem of having important boundary effects that perturb the trajectories?

I noticed only very few and harmless typos in the text that the authors will probably already have corrected or will correct when rewriting part of the paper during the review process. However, there were things missing or wrong in some figure legends - authors should check them and improve their clarity. For instance:

- what is the difference between circles and diamonds in fig 2?
- a instead of A in the legend of fig 4.
- in supp. fig. 5, the concentration of aspartate for a-c is not specified (should be 200 micrograms)
- supp fig. 6a in the pink curve legend should be diff. coeff. μ

There is a typo in equation 8. Either there should be no index i in ρ , or if you want different phenotypes to produce different amounts of S , the α_S should also depend on i and the terms should be summed in i .

We thank the reviewers for their detailed remarks on our original manuscript, which were extremely helpful. We have tried to address all of their comments and concerns, which entailed a number of new experiments, data analysis, and modeling. The result is, we believe, a significantly improved paper. The main issues raised by the reviewers are collected together below, with a summary of how we dealt with them. This is followed by detailed responses to each of the reviews.

Concerns about cell labeling and channel depth: The new Extended Data Figure 2 shows that labeling cells with fluorescent proteins, or changing the depth of the microfluidics channel from 14 μm to 100 μm does not affect tumble bias distribution nor wave speed.

Concerns about the lifetime of the waves in our experiments and of the spatial sorting within them: The new panels of Extended Data Figure 5 show that after an initial transient, the structure of the traveling wave and the number of cells in the wave are stable, with only a slow decay. Moreover, the new Fig. 4e and Extended Data Fig 9d show that the spatial arrangement of phenotypes is stable as well (over more than 3000 s in Extended Data Fig 9d using a much longer channel). Finally, the new Fig 4f confirms, with a many more experiments than before, that the distance between phenotypes within one traveling wave increases with the difference in tumble bias. This new data demonstrates that the phenomena described are not short transients.

Lack of direct evidence of oxygen depletion in the middle of the wave: Following the reviewers' suggestion, Extended Data Figure 7 now shows direct evidence that dissolved oxygen is reduced in the middle of the traveling wave where cell density is high, confirming our previous indirect evidences. The data was obtained by synthesizing the fluorescent reporter from Douarache et al., mixing it with the cells, and imaging the fluorescence signal as the band travelled. We thank the reviewers for this excellent suggestion that strengthens our paper.

Bring the mathematical model out of the SI and plot predictions on top of the data: Although this is first and foremost an experimental study, modeling was used to generate a mechanistic hypothesis that was then tested experimentally. Accordingly, the model was derived by making the simplest possible modification to the Keller-Segel model to account for a continuous distribution of phenotypes and oxygen consumption. Model predictions are now plotted on top of the data in Figure 2, showing good agreement. The model is now described in the main text (first section of Methods).

Placing the new results in the context of previous work: We added text in the introduction and conclusion to better place our work in the context of previous work and to clearly separate the two results of our paper: the self-sorting mechanism, which helps compensate for differences in chemotactic abilities, and the role of oxygen consumption, which helps reduce the leakage of high tumble bias cells at the back of the wave.

Reviewer #1 (Remarks to the Author):

This manuscript by Fu and colleagues uses bacterial chemotaxis as a model system to ask a general question: how can organisms migrate at a group despite differences between individuals in the ability to move? This manuscript builds on previous work by this lab and others that recorded marked differences between individual bacterial cells in the ability to chemotax in gradients of attractants. At least part of these differences between individuals are based on variation in the 'tumbling bias' – the frequency at which cells interrupt runs by tumbles. Tumbling bias directly translates to a cell's ability to swim up a gradient of an attractant. One might thus expect that cells with different tumbling biases would become separated from each other during chemotaxis in a gradient. However, and intriguingly, this is not the case; despite the variability in chemotaxis performance, these cells migrate as a group. The paper describes how this works, and thereby provides a new and fundamental insight into the link between

individuality and collective behavior.

The main discovery put forward by the authors is that the individuals with the strongest chemotactic performance change the local environment (the gradient of the chemoattractant) in a way that allows individuals with lower performance to keep up with the rest. A number of quantitative single-cell experiments support the main aspects of this idea: the authors construct different mutants that vary in their chemotactic performance and show that these mutants can migrate together; and they show that the spatial organization of the mutants corresponds to their prediction, with the types with the strongest performance traveling at the front and the types with weaker performance further back (but still as part of the same group, defined as a moving unimodal distribution of bacterial density along the gradient.)

This manuscript addresses a fundamental question ('how does collective behavior emerge in heterogeneous groups'), it puts forward an intriguing new concept, and supports it with conclusive experimental data. I thus think that the paper will make a seminal contribution, and be of interest to scientists broadly interested in collective behavior, complex biological systems, and to scientist interested in the microbial sciences.

We are pleased that the reviewer found our results important and of broad interest.

I have a few of suggestions and comments.

3th paragraph discussion ("Because the range of phenotypes ...")

- The phrasing "spatially allocates labor" can be misinterpreted to suggests that cells in different parts of the wave perform different tasks (as in division of labor), however the authors showed that all cells perform the same task, but perform this tasks with different proficiency. I suggest to rephrase this sentence.

We have rephrased as follows: "Because the range of phenotypes allowed within a travelling group depends on the spatial profile of the traveling gradient, this mechanism introduces important feedbacks between the environment, cellular metabolism, and phenotypic diversity, which together generate spatial patterns of phenotypes according to functional capabilities."

- "...suggest a role for non-genetic diversity in generating functional structure within cellular communities." This phrasing suggests that by sorting cells with different chemotactic performance in the migration wave new functionality is achieved at the group level. It is not clear to me whether this is the right formulation. Without phenotypic heterogeneity cells can form migrating bands. Phenotypic heterogeneity could potentially prevent these bands from forming, but the sorting described by the authors provides a solution to this problem. I think it is thus fair to say that sorting allows functionality at the group level in-spite of the potential debilitating effects of heterogeneity, but I would not call this new functionality. I would suggest to rephrase this sentence.

We have removed this sentence and made sure to use consistent phrasing throughout. We have also spent more of the Discussion text distinguishing the two results of the paper: self-sorting mechanism and oxygen-dependent effect.

- "In the presence of genetic variation ...". It is not clear to me what exactly the authors mean with this statement and how it is related to the findings shown in the manuscript. I suggest to remove or clarify this statement.

We have removed this sentence.

Fig 2

“...the cell density profile became narrower (Fig. 2d)”, page 5

The large difference in the maximum cell density between conditions makes it hard to compare the distributions in Fig. 2d to validate this statement. It would be helpful to have a quantification of the width of the distribution or to show normalized graphs of the density profile.

Following the reviewer’s suggestion, we added normalized cell density profiles (pdfs) in Extended Data Fig. 5.

“Here, we assume the self-attractant to be glutamate...” Page 14 SI

Why did the authors choose glutamate? Is this a conservative assumption? Or can the authors give a rough estimate of how sensitive their conclusions are to the choice of the K_0 and K_1 values? I don’t think that a full sensitivity analysis is needed, but I think it would be useful for readers to get an idea of how strongly the conclusions depend on this assumption.

We had used glutamate as the self-attractant because according to our HPLC analysis, of all the amino acid metabolic products secreted by *E. coli* upon consumption of aspartate, glutamate was the one that could have had the largest effect, albeit still much smaller than the response to aspartate. Unfortunately, in our original manuscript we did not clearly explain the multiple experiments (including HPLC) we conducted to check for the presence of potential self-attractant or the potential role of aerotaxis (see new Extended Data Figure 6). We now mention these experiments in the main text and have added a paragraph in the Methods explaining our HPLC experiments more thoroughly, showing that amino acids and oxygen can be neglected as possible attractants compared to aspartate. Based on these experiments and the fact that we now have direct evidence of oxygen limitation in the wave, we decided not to include simulations in this paper that explore known past results, such as how a strong self-attractant can help reduce cell leakage. Instead, we cite past works that make this point and focus on illustrating in Extended Data Figure 8b how the oxygen-dependent effect also helps reduce cell leakage.

“...slow decay due to cells falling off at the back of the band”, page 5

In several places the authors mention that densities in the band decrease due to cells falling off the back of the wave. Did the authors also consider the possibility that some cells escape at the front (i.e. run ahead of the wave)? Can the relative importance of these two possibilities be estimated, or is there an a-priori reason to assume that most loss is at the back?

Cells fall off the back rather than run ahead of the band because there is no gradient ahead of the band for cells to climb, whereas behind the band it is possible for cells moving at drift velocities less than the band speed to be left behind. Cells at the front can diffuse ahead of the wave, but the distance covered via diffusion grows only as the square root of time, whereas the wave covers distance linearly in time. Thus, the wave catches up with cells that diffuse ahead.

Minor comments:

Fig 2b,c:

The difference between the diamond and circle symbols is not explained in the caption.

We added an explanation of the meaning of diamonds and circles in the caption. circles: experiments in which tumble bias were measured and used in e and f; diamonds: experiments in which tumble bias were not measured.

Extended Data Fig 6e,f,

To facilitate comparison of data and model it would be very helpful to add the experimental data to these panels.

We have placed the model prediction on top of the data as dashed lines in Fig 2. The resulting model (Methods and Table 1) qualitatively reproduces the main features of our experiments, including the dependency on aspartate concentration of the band speed (Fig. 2b), cell number and density (Fig. 2cd), tumble bias distribution (Fig 2e), phenotypic selection (Fig. 2f), and average aspartate consumption rate per cell as a function of cell number in the band (Extended Figure Data 7h). In Extended Data Figure 8b we show how the oxygen-dependent effect helps reduce cell leakage.

Extended Data Fig 8b,c:

The caption does not include a description of the meaning of the colors of the lines

We have added an explanation in the legend.

Reviewer #2 (Remarks to the Author):

This article investigates the impact of diversity in collective chemotaxis of bacteria. It combines experimental observations and theoretical analysis. This is an interesting idea that has been previously explored (see in particular Emako et al. PLoS Comp Biol (2016)).

To better place our study in the context of past works, we have added text in the Introduction and Conclusion. Although the migration of traveling waves or “bands” of bacteria along a channel has also served as a classic model for the theoretical study of emergent phenomena and pattern formation in biology, the effect of non-genetic diversity on this process has scarcely been examined. Previous studies (Lin, T. C., & Wang, Z. A. 2014. *Discrete and Continuous Dynamical Systems*, 34(7), 2907–2927 and Emako et al. 2016) examined how two populations may travel together. However, these studies assumed that within each population all the individuals were identical. The mechanisms by which a continuum of phenotypes, known to always be present in bacterial populations (Spudich and Koshland *Nature* 1976; Waite et al. *MSB* 2016), can achieve coherent migration have not been investigated. Our study addresses this question by characterizing the phenotypic heterogeneity of the individual cells that traveled in the band, emphasizing the fact that there is a continuous distribution of swimming phenotypes there. The sorting mechanism and the oxygen-dependent effect reported here are novel.

The experiments are very carefully performed and the collected data appears to be of high quality. For the sake of clarity, I suggest to move the set of experiments performed in presence of casamino acids that remains very qualitative, to the Supplementary Material. The core of the paper deals with the more controlled experiments performed in presence of aspartate only and their interpretation by the proposed theoretical model.

The experiments performed in the presence of casamino acids have important biological significance, and for this reason we think they should remain in the main text. Fig 1 demonstrates one of our key findings: that collective migration puts bounds on the amount of diversity in the group and that this selection is non-genetic, since upon growth the original distribution of phenotype is recuperated. These experiments further show that this selection process is general, since it applies also to the more natural case of an undefined medium. Finally, Fig. 1 also show that population behavior in our experiments is similar to that demonstrated early on by Julius Adler (Adler, J. *Chemotaxis in bacteria*. 1966 *Science* **153**, 708-716): in undefined media, such as casamino acids, several bands emerge because cells consume different compounds at different rates.

On the experimental side, more details on the microfluidic setup should be given. The fabrication of the device could be made more explicit with a schematic.

Following the reviewer’s suggestion, we added a schematic about the microfluidics fabrication in Extended Data Figure 1a and referred to it in the Methods.

If I understand correctly, the depth of the PDMS channel is only 10 μm (smaller than a run length). How can the authors then rule out the influence of the surfaces that has been evidenced in many studies? This small depth may be related to the large pulse velocities (larger than 10 $\mu\text{m/s}$).

To address the reviewer’s concern, we have performed new migration experiments in a straight channel of 100- μm depth. In 200 μM aspartate, we measured average band speeds of 0.18 ± 0.03 mm/min ($n = 7$), which is essentially identical to that measured in the shallow device used in the paper (Fig. 2b, 0.18 ± 0.01 mm/min) (\pm indicates standard deviation) (Extended Data Figure 2b). Thus, the depth of our device does not strongly affect the band speed. Importantly, here we used the same channel depth, the same strain (RP437), and the same motility buffer as in our previous publications so that our new results can be

interpreted in a consistent manner in the context of the information about phenotypic heterogeneity gathered in these previous studies.

Surprisingly, the band speeds and densities are measured with mixtures of labeled and unlabeled cells (p5), although the reason for that is not clear. This is problematic as the emphasis of the paper is put on having a pure clonal population. Fig. 2C shows that the number of fluorescent cells increases with [Asp]. It is true of the total number? In the same proportion?

Various groups before us have used mixtures of labeled and unlabeled cells in such experiments (e.g. Mittal et al. 2011 PNAS, 100(23), 13259–13263; Saragosti et al. 2011 PNAS, 108(39), 16235–16240). As a control, we show in a new Extended Data Figure 2 that labeling cells with mRFP1 or YFP does not affect the tumble bias distribution nor band speed.

Since the experiments basically confirm previously published results, the true originality of the paper is the theoretical model. The model is based on a modified version of the Keller Segel model that includes the concentration of oxygen in the medium and a dependency of aspartate consumption with oxygen concentration.

We respectfully disagree. Our experiments characterize for the first time both the distribution of individual cell phenotypes within the band and the macroscopic properties of the band, such as speed, cell density, and number of cells, all in the same experiment. In doing so, these experiments enable us to examine how a continuum of phenotypes, known to always be present in bacterial populations, can achieve coherent migration. Although this is first and foremost an experimental study, modeling was used to generate a mechanistic hypothesis that was then tested experimentally. The key modification here of the Keller-Segel model is the inclusion of a continuous distribution of phenotypes in the population, which leads to spatial sorting of phenotypes within the traveling band.

This is a non-trivial component as experiments are performed in PDMS microchannels, a material very permeable to Oxygen, in very shallow channels. As mentioned by the authors themselves, a more common way to write these KS equations is to involve an attractant secreted by the bacteria. Some arguments in favor of the importance of oxygen consumption vs attractant secretion are given in the Supplementary Notes. However, they remain very qualitative. Besides, Douarche et al. PRL (2009) have shown that the diffusion coefficient of the bacteria is a highly non-linear function of oxygen concentration. Shouldn't this effect or simply aerotaxis be then included in the equations? Since the model is the truly original part of the paper, it appears absolutely critical to directly measure the local oxygen concentration in the channel (Douarche et al. PRL 2009) to validate the different hypotheses in a more direct way than the present indirect experiments listed in Supp Fig 7.

We thank the reviewer for the interesting suggestion to try measuring oxygen *in situ*. We have done so (Extended Data Figure 7) and found that oxygen is indeed reduced in the middle of the band where cell density is high (see response to Reviewer 3). We also checked that aerotaxis was not essential for band migration using a mutant lacking the oxygen receptor *aer*- (Extended Data Figure 6d).

Regarding the dependency of the diffusion coefficient on oxygen. In Douarche et al., the diffusion coefficient is almost a step function of oxygen: the curve is flat and drops rapidly to zero when the partial pressure of oxygen drops below a threshold near zero. By inspection under the phase contrast microscope, we can see that all of the cells in the wave are motile. This suggests that the oxygen levels in the wave does not drop below the threshold in Douarche et al, and the diffusion coefficient can be considered constant.

The dependence of the diffusion coefficient with bacterial density (Wu et al. PRL 2000) and Saragosti PNAS (2011)) should also be included.

Saragosti et al. PNAS 2011 determined that the dependence of the diffusion coefficient on bacterial density was negligible in the band.

As the article stands now, the comparison between theory and experiments remains too qualitative and the proposed model describes the experiments no worse or better than other models that include chemoattractant secretion. It is surprising that no figure compares directly and quantitatively experiments and simulations. In the table recapitulating the results, many parameters are not explicitly given.

In Extended Data Figure 6, we used quantitative HPLC to measure the amount of amino acids that cells secrete upon consuming aspartate as a function of time. Our measurements demonstrate that aspartate dominates the response to amino acids in our experiments (Methods and Extended Data Figure 6c). Thus, although we cannot rule out that some undetected self-attractant may be secreted, we also cannot justify including secretion of a self-attractant in the model. Moreover, thanks to the reviewer's suggestion to measure oxygen in situ, we now have direct evidences for oxygen limitation within the wave.

Following the reviewer's request, we have placed the model predictions on top of the data as dashed lines in Fig 2.

We apologize for the missed parameters and have completed the table.

Moreover, the 4 strains engineered to have different TBs are not characterized independently. Why isn't the behavior of each of these strains studied separately (pulse velocities, diffusion coefficients, chemotaxis coefficients), and compared to theory? How does the peak velocity of a given strain alone compare with the one it acquires when mixed with another one? Looking closely at Fig. 4b, it looks like the two bands travel at slightly different velocities and therefore may end up separated if one waits sufficiently long. In other words, the distance between the peaks in Fig. 4c is itself an increasing function of time. It would be useful to add the kymographs of these experiments at longer times. At what time point were the Fig. 4c, 4E or 4F measured? What did guide the choice of this particular time?

The new Fig. 4e and Extended Data Fig 9d show that the spatial arrangement of phenotypes is stable (over more than 3000 s in Extended Data Fig 9d where we used a much longer channel). This new data demonstrates that the phenomena described are not short transients. In all of the figures showing cell density profiles, we now report the time intervals between independent measurements.

In a similar note, Fig 3b and 4f don't seem to fit: the distance between peaks is larger in the simulations compared to the experiments. Again, a direct comparison would be welcome.

As mentioned above, we have now put all simulation predictions on top of the data in Fig. 2. Fig 3b and 4f cannot be directly compared because in 4f we are mixing two populations each with its own distribution of TB, whereas the simulation in 3b is for a population with wild-type distribution of TB. Finally, we have redone Fig 4f. conducting a large number of new experiments to increase the number of data points in Fig. 4f from 2 to 8 (each data point requiring n=4 independent experiments). The result clearly shows how physical distance within the traveling wave increases with phenotypic distance (difference between the mean TB of the two populations).

Finally, would it be possible to use gated channels as described in the first part to extract the tumble bias

of each population while in the wave, or to obtain this information by analyzing the trajectories of the bacteria in situ (within the wave)?

Within the traveling band the cells are responding to the traveling gradient, which can be fairly steep. Therefore, the tumble bias of a cell is continuously changing depending on its direction of motion within the gradient. While it would be interesting to analyze trajectories of bacteria in situ, our goal here was to determine the distribution of phenotypes that migrate in a traveling band. For that, it is necessary to measure the “intrinsic” tumble bias of individual cells in the absence of a gradient. We did that using the gates on our device to capture the cells from a single band inside the large observation chamber after they had traveled together.

Reviewer #3 (Remarks to the Author):

In this work, the authors study a clonal population of motile E. coli cells that undergo collective migration in a channel filled with nutrients. This collective behavior is fueled by the fact that the bacteria consume the nutrient generating a chemical gradient of nutrient concentration to which they respond. Using a nicely conceived microfluidics device, the authors can obtain measurements of a non-homogeneity of the tumbling bias of the bacteria at different positions within the traveling band. Based on their observations the authors propose a model to justify how bacterial populations with such a heterogeneous phenotype are able to migrate collectively.

The mechanism proposed in this work for keeping bacteria of different tumbling bias swimming together in the same band instead of falling apart is original and very interesting. Indeed, this procedure enables bacteria to achieve this without a chemotactic substance produced by the bacteria themselves and to which they respond giving them a natural way of keeping together which was an essential component of the mechanisms described in many previous works).

We are pleased that the reviewer found the new mechanism original and important.

Even when there is a self-attraction term, traveling together is not granted and depends on more intricate phenomena : it had even been shown that in case of a traveling speed phenotype difference, bacteria could travel as a single pulse or split into two separate pulses depending on their relative abundance in the population (see reference [9] of the supp. material). This gives a great value to having been able, in the present work, to propose a completely different model that produces collective behavior without the need of a self-attractant and which is coherent with the experimental observations obtained concerning the spatially heterogeneous tumbling bias phenotype inside the traveling band.

We added text in the introduction and conclusion to better place our work in the context of previous work and also to clearly separate the two results of the paper: the self-sorting mechanism, which helps compensate for differences in chemotactic abilities, and the role of oxygen consumption, which helps reduce the leakage of high tumble bias cells at the back of the wave.

However, the experimental evidence for certain parts of this model, in particular the role of oxygen consumption, is rather circumstantial and I think that the value of this paper would be greatly improved if the authors could produce direct evidence that the actual oxygen distribution in the channel agrees with that predicted by their model. There exist fluorescent markers for oxygen described in the literature (see, for instance, C. Douarche et al. PRL 102, 198101 (2009)). Having the same type of information concerning space and time dependence of aspartate concentration would also be nice to check equation 2 of the model (in particular how the aspartate consumption depends on oxygen), but this should be harder.

We thank the reviewer, as well as reviewer 2, for this suggestion. We contacted the authors of the Douarche et al. paper and received the protocol for synthesizing the oxygen sensor. However, it was pointed out to us that the high density of cells in the wave could partially exclude the dye, creating variations in fluorescence intensity in the band that were not entirely due to differences in oxygen concentration. This makes quantitatively measuring the oxygen concentration in the band very difficult. Nevertheless, we synthesized ruthenium-micelles (the dye must be encapsulated in lipids to avoid toxicity to the cells) and used them to monitor dissolved oxygen in situ. In 100-um deep channels, we were able to clearly detect a profile of fluorescence traveling with the band of cells, indicating that oxygen was depleted in the middle of the traveling wave (Extended Data Figure 7ab). In the shallow devices, the signal was much weaker and noisier but also traveled with the band (Extended Data Figure 7cd). To increase the signal to noise ratio, we tried increasing the concentration of micelles loaded, but this resulted in traveling waves that stopped their migration because the cells became non-motile, possibly

because of a lack of oxygen, sequestered by the micelles. Taken together these measurements show that oxygen availability is reduced in the center of the wave and support our earlier, indirect evidence that the average aspartate consumption rate per cell depends on the cell density in the band (Methods Eq. (4), Extended Data Figure 7h).

Another important point that is not very clear in the paper and would deserve being clarified is if the traveling bands described in this paper are a transient phenomenon or if they are intrinsically permanent in ideal conditions (with or without considering proliferation for longer time scales). In fact, on the one hand the loss of mass in the peaks seems to not be limited to the grey areas (see fig. 3d) and on the other the propagation speed seems to decrease (visual impression from fig. 2a and supp. fig. 7). In general, we have nearly exclusively data about distribution profiles corresponding to snapshots (for which the time is sometimes not given) and very little about how things change in time. It would be important to give more information on the evolution in general. For instance how far are we from the idealized situation of a peak of constant mass traveling at constant speed? (which is an important issue for the choice of appropriate mathematical models).

As shown using light scattering by Holz and Chen (1978 *Biophysj*, 23, 15) and theoretically by Novick-Cohen & Segel (1984 *J of Math Bio*, 19, 125), as well as more recently by Saragosti et al (PNAS 2011), the finite sensitivity of cells to attractants always leads to a slow decay of the traveling wave, unless the number of cells in the wave is replenished by growth. In our experiments, the same is true. To address the reviewer's concern, the new panels of Extended Data Fig 5 show that after an initial transient, the structure of the traveling wave and the number of cells in the wave are stable, with only a slow decay. Moreover, the new Fig 4e and Extended Data Fig 9d show that the spatial arrangement of phenotypes is stable as well (over more than 3000 s in Extended Data Fig 9d using a much longer channel). Finally, the new Fig 4f confirms, with a many more experiments than before, that the distance between phenotypes within one traveling wave increases with the difference in tumble bias. This new data demonstrates that the phenomena described are not short transients. In all of the figures showing cell density profiles, we now report the time intervals between independent measurements. Note that we are plotting time-averages of the cell density in each case.

If I understood properly, in supplementary figure 6f the authors just indicate that their simulations predict an important density decrease if there is no oxygen dependence on aspartate consumption. What about the experimental results?

Conducting the same types of experiments in the absence of oxygen could have effects not related to aspartate consumption because the cells would become anaerobic, which would make comparison to our experiments difficult. In addition to the simulations shown now in Extended Data Fig. 8b, we have added an analytical derivation explaining how oxygen-dependence reduces cell leakage (Methods, Eq. (9)).

One more issue that worries me concerning the data obtained using fluorescently labeled cells to deduce general properties of unlabeled populations (and which is not specific to this work) is that the results presented in supp. reference [9] showed that two e.coli cell populations, where the only difference was the fluorescent label, had very different physical behaviors (in particular different pulse speeds in a similar channel migration experiment). Even if the results obtained seem to be coherent with the assumption that the fluorescent labeling has little or no influence, it would be important for the authors to check carefully what are the consequences of the labeling in their experimental setting.

We have conducted new control experiments showing that labeling cells with either YFP or mRFP1 does not affect tumble bias distribution nor band speed (Extended Data Figure 2).

As a general remark, I find it unfortunate that the mathematical contents are nearly all "hidden" in the supplementary material. In this paper, the mathematical model is crucial for giving impact to the experimental observations - the observation of the spatial structuring of the band in tumbling bias phenotype without the plausible explanation given by the mathematical modeling would have considerably less value. Considering its importance in this study, the mathematical material should be included in the main paper, at least in the methods section (as the same group did in supp. ref. [6]). And some more important aspects that single out the contribution of this work relative to previous mathematical studies, deserve to have a place in the main text, even if they are not always appealing for biological readers. This is also important for them to progressively learn how to appreciate mathematical modeling contributions.

Following the reviewer's advice, we have moved the mathematical model and the derivations of the sorting rule and the effect of oxygen-dependent consumption to the Methods section of the main text.

To end, just a couple of short remarks:

- I was puzzled by the precision of the expressions in equations 4 and 5 of the supp material when we see the degree of variability of the experimental results (like the one we see in figure 4f for the mean tumble bias difference of 0.3).

We have simplified these equations to reduce the complexity of the model, while still maintaining the essential dependencies of μ and χ on tumble bias observed in Waite et al. (2016 MSB 12:896,1-14) using the same strain (RP437), similar motility buffer, and same microfluidics depth as used here.

The original Figure 4f was not plotted correctly. All data points for the same inducer level were plotted at the same position on the horizontal axis. But small experimental variations in inducer levels are unavoidable and lead to differences in mean tumble bias, which should be taken into account when plotting the data. We have corrected this plot and have conducted a large number of new experiments to increase the number of data points in Fig. 4f from 2 to 8 (each data point requiring n=4 independent experiments). The result clearly shows how physical distance within the traveling wave increases with phenotypic distance (difference between the mean TB of the two populations).

- Doesn't having a 10 micrometer thickness, which is not big compared to the mean run distance, pose a problem of having important boundary effects that perturb the trajectories?

Please see our answer to the same question by reviewer 2.

I noticed only very few and harmless typos in the text that the authors will probably already have corrected or will correct when rewriting part of the paper during the review process. However, there were things missing or wrong in some figure legends - authors should check them and improve their clarity. For instance:

- what is the difference between circles and diamonds in fig 2?

We have added captions in Fig. 2 to explain the circles and diamonds.

- a instead of A in the legend of fig 4.

Corrected.

- in supp. fig. 5, the concentration of aspartate for a-c is not specified (should be 200 micrograms)

This has been added.

-supp fig. 6a in the pink curve legend should be diff. coeff. μ

We have corrected this typo.

There is a typo in equation 8. Either there should be no index i in ρ , or if you want different phenotypes to produce different amounts of S , the α_S should also depend on i and the terms should be summed in i .

We have removed the self-attractant from our simulations (see response to reviewer 1).

Reviewers' comments:

Reviewer #1 (Remarks to the Author):

All my comments have been addressed in the revised version, and I have no additional comments.

Reviewer #2 (Remarks to the Author):

The authors have made substantial modifications to the first version of their manuscript. I find the article more balanced and the questions better posed. In particular, dealing with the role of oxygen, I commend the authors for having performed these new experiments that support their model.

However, I still have difficulties with some parts of this paper.

1/ All the wave experiments reported in the literature necessarily encompass the diversity outlined by the authors. How do the experiments and conclusions of the present article differ from say Saragosti et al. (2011) who have also observed that cells at the back and front of the wave have different swimming characteristics? How do these two papers compare quantitatively? Such quantitative comparison should be included in the discussion.

2/ The authors note at several places that the present article is a significant advance because it considers a continuum of phenotypes and not two phenotypes only as in previous studies. Indeed the theory deals with a discrete sum of phenotypes; however, when it comes to testing the theoretical predictions experimentally, this is performed with two phenotypes only.

3/ How stable in time are the strains corresponding to the 4 phenotypes depicted in Figure 4a, in the conditions used for the wave experiments (during which they experience the same level of aTc)?

4/ These 4 phenotypes differing in their TBs are not individually characterized for their ability to make waves. What are the speed and dispersion of these waves? Their characteristics should be presented for all the 4 phenotypes. Shouldn't the knowledge of these characteristics when the strains are taken separately, be sufficient to quantitatively predict their behavior when they are mixed?

5/ The theoretical curve in Fig 3d appears to be bell-shaped with a maximum around the peak ($z=0$). What is the meaning of the decrease of this curve at the front of the wave (visible at $z\sim 0.5\text{mm}$)? The horizontal scale in this particular panel should go from $z=-1.5\text{mm}$ to $z=+1\text{mm}$ as the other panels. The vertical scale should be adapted to the data.

6/ Similarly, Figure 4f is plotted for small TBs/distance between peaks; it should be expanded to larger values by including all experiments (for instance the point corresponding to Fig 4c corresponds to an inter-peak distance of typically 0.8 mm for a difference of TBs of typically 0.4).

7/ As mentioned in the first review, the diffusion coefficient is proportional to the bacterial density (Wu et al. 2000, Saragosti et al. 2011). What are the consequences of this relationship in the theoretical analysis?

Generally speaking, I strongly recommend to the author to discuss more closely and more quantitatively their results at the light of the existing literature.

Minor point: there is a problem with the colors of my version of Figure S1a.

Reviewer #3 (Remarks to the Author):

The authors made a significant revision that improves their work. In particular, it answers most of the concerns that I had expressed. I am sorry that, due to technical difficulties the authors were not able to have further direct information concerning the oxygen consumption but this is well justified by the authors. In my opinion this new version deserves being published in Nature Communications.

We thank the reviewers for reading our revised manuscript in details. We are glad that reviewers 1 and 3 have found our revision to be substantial and to address all of their concerns. Below we address the remaining comments of reviewer 2.

Reviewer #1 (Remarks to the Author):

All my comments have been addressed in the revised version, and I have no additional comments.

Reviewer #3 (Remarks to the Author):

The authors made a significant revision that improves their work. In particular, it answers most of the concerns that I had expressed. I am sorry that, due to technical difficulties the authors were not able to have further direct information concerning the oxygen consumption but this is well justified by the authors. In my opinion this new version deserves being published in Nature Communications.

Reviewer #2 (Remarks to the Author):

The authors have made substantial modifications to the first version of their manuscript. I find the article more balanced and the questions better posed. In particular, dealing with the role of oxygen, I commend the authors for having performed these new experiments that support their model.

We are glad that the reviewer appreciated the new data.

However, I still have difficulties with some parts of this paper.

1/ All the wave experiments reported in the literature necessarily encompass the diversity outlined by the authors. How do the experiments and conclusions of the present article differ from say Saragosti et al. (2011) who have also observed that cells at the back and front of the wave have different swimming characteristics? How do these two papers compare quantitatively? Such quantitative comparison should be included in the discussion.

We added a new paragraph in the discussion to address the distinction between dynamic (extrinsic) and phenotypic (intrinsic) differences and explain how they contribute to spatial structure in the traveling group. We also added a paragraph to compare our results about the migration of a population made of a continuum of phenotypes with previous models of the migration of two phenotypes.

2/ The authors note at several places that the present article is a significant advance because it considers a continuum of phenotypes and not two phenotypes only as in previous studies. Indeed the theory deals with a discrete sum of phenotypes; however, when it comes to testing the theoretical predictions experimentally, this is performed with two phenotypes only.

There is an important distinction between populations and phenotypes in our study that is not made in previous studies. We do not test our theoretical predictions with only two phenotypes. Experimentally it is not possible to generate a population of a single phenotype or to label individual phenotypes within a population. For this reason, we test our predictions by using *multiple* pairs of *populations* (labeled with different fluorescent proteins). Each population consists of an entire *distribution* of phenotypes, and we change systematically the differences between the means of these distributions over a wide range to generate mixed populations.

3/ How stable in time are the strains corresponding to the 4 phenotypes depicted in Figure 4a, in the conditions used for the wave experiments (during which they experience the same level of aTc)?

In the main text we did not make clear that aTc was only used during cell growth. aTc was washed away with the growth media when cells were resuspended into buffer before starting the wave experiment. Thus, the wave experiments shown in Figure 4 took place in absence of aTc and there is no further expression of CheZ. We added a note in the main text to mention this clearly. In buffer, the cells do not grow and the tumble bias distribution is stable for at least one hour (Dufour et al. PLoS Comp Bio 2016).

4/ These 4 phenotypes differing in their TBs are not individually characterized for their ability to make waves. What are the speed and dispersion of these waves? Their characteristics should be presented for all the 4 phenotypes. Shouldn't the knowledge of these characteristics when the strains are taken separately, be sufficient to quantitatively predict their behavior when they are mixed?

As we mention above, these are not phenotypes, they are populations with entire distributions of phenotypes. We do not think that simply knowing the speed of each individual population is enough to predict the speed of the mixture between two populations. The speed of the mixture depends on the number of cells able to migrate, which is a complex function of initial conditions that operates on the entire mixture distribution of phenotypes. Studying this would be the subject of another study and is outside the scope of this paper. In our paper we use these mixtures of distributions of phenotypes only to experimentally demonstrate the spatial sorting.

5/ The theoretical curve in Fig 3d appears to be bell-shaped with a maximum around the peak ($z=0$). What is the meaning of the decrease of this curve at the front of the wave (visible at $z\sim 0.5\text{mm}$)? The horizontal scale in this particular panel should go from $z=-1.5\text{mm}$ to $z=+1\text{mm}$ as the other panels. The vertical scale should be adapted to the data.

Thank you for catching this inconsistency in the horizontal axes of the plots in figure 3. We have extended the horizontal axis in Fig 3d to match the other panels. The rightmost points in Fig. 3c (blue line) and Fig. 3d (black line) correspond to the phenotype with the maximum chemotactic coefficient in the band. Ahead of that relative location, there are no more peaks in the cell density of any phenotype, however there are cells due to diffusion. We now explain this in the main text. The tiny dip at the right end of the black line in Fig 3d was due to the finite spatial resolution in our simulations: the peak of the density profile of one phenotype is not necessary aligned with a grid point. We reran the simulation at higher resolution and replaced the line in the figure. The tiny dip is now gone.

6/ Similarly, Figure 4f is plotted for small TBs/distance between peaks; it should be expended to larger values by including all experiments (for instance the point corresponding to Fig 4c corresponds to an inter-peak distance of typically 0.8 mm for a difference of TBs of typically 0.4).

The experiments in Figure 4a-e used four different strains (fluorescent and non-fluorescent cells, each at two inductions levels) so that the density profiles could be measured. Mixing populations with closer TB distributions caused the peaks of the two traveling populations to be closer (magenta and green in Figs. 4ad and Extended Data Fig 9c for replicates), suggesting that distance between peaks increases with difference in $\langle TB \rangle$ as a result of spatial sorting. Due to experimental limitations, we could not measure the TB distributions of all four strains in each of the experiments reported in Figs. 4a-e. To better quantify the relationship between peak separation and difference in $\langle TB \rangle$ value, we instead mixed pairs of populations using only fluorescent strains induced with different levels of aTc. For each pair, we measured the TB distributions right before loading cells in the device, and then measured the distance between the fluorescence peaks in the resulting traveling band (Fig 4f). This confirmed that there is a monotonic

relationship between peak separation and difference in $\langle TB \rangle$ values. Due to the difference in experimental protocols, we have not included the data in Figure 4a-e in Figure 4f. We rephrased the main text describing Fig 4f to make this difference clear.

7/ As mentioned in the first review, the diffusion coefficient is proportional to the bacterial density (Wu et al. 2000, Saragosti et al. 2011). What are the consequences of this relationship in the theoretical analysis?

Saragosti et al. PNAS 2011 determined experimentally that the dependence of the diffusion coefficient on bacterial density had a negligible influence on the propagation of the band. In our experiments cell density is similar to that in Saragosti's experiments. Therefore, we followed Saragosti's conclusion and made the same assumption in our model. We mentioned this in the main text.

Generally speaking, I strongly recommend to the author to discuss more closely and more quantitatively their results at the light of the existing literature.

Following the reviewer's suggestion we added two new paragraphs in the Discussion to better compare our results with previous results in the literature.

Minor point: there is a problem with the colors of my version of Figure S1a.

Fixed. This was a problem with the rendering of the image into the PDF file.

REVIEWERS' COMMENTS:

Reviewer #2 (Remarks to the Author):

The authors have now answered most of my concerns. I think that this article can be published. To improve clarity, I think it should be mentioned that the TB distribution remains stable for 1 h after having washed out the inducer (point 3 of the preceding review).

REVIEWERS' COMMENTS:

Reviewer #2 (Remarks to the Author):

The authors have now answered most of my concerns. I think that this article can be published. To improve clarity, I think it should be mentioned that the TB distribution remains stable for 1 h after having washed out the inducer (point 3 of the preceding review).

We have added a sentence to the Results to clarify this point.